# Efferocytosis requires periphagosomal Ca$^{2+}$-signaling and TRPM7-mediated electrical activity

Michael S. Schappe [1,4], Marta E. Stremska[1], Gregory W. Busey [1], Taylor K. Downs[1], Philip V. Seegren[1], Suresh K. Mendu[1], Zachary Flegal[1], Catherine A. Doyle [1], Eric J. Stipes [1] & Bimal N. Desai [1,2,3]✉

Efficient clearance of apoptotic cells by phagocytosis, also known as efferocytosis, is fundamental to developmental biology, organ physiology, and immunology. Macrophages use multiple mechanisms to detect and engulf apoptotic cells, but the signaling pathways that regulate the digestion of the apoptotic cell cargo, such as the dynamic Ca$^{2+}$ signals, are poorly understood. Using an siRNA screen, we identify TRPM7 as a Ca$^{2+}$-conducting ion channel essential for phagosome maturation during efferocytosis. *Trpm7*-targeted macrophages fail to fully acidify or digest their phagosomal cargo in the absence of TRPM7. Through perforated patch electrophysiology, we demonstrate that TRPM7 mediates a pH-activated cationic current necessary to sustain phagosomal acidification. Using mice expressing a genetically-encoded Ca$^{2+}$ sensor, we observe that phagosome maturation requires peri-phagosomal Ca$^{2+}$-signals dependent on TRPM7. Overall, we reveal TRPM7 as a central regulator of phagosome maturation during macrophage efferocytosis.

[1] Pharmacology Department, University of Virginia, Pinn Hall, 1340 Jefferson Park Avenue, Charlottesville, VA 22908, USA. [2] Carter Immunology Center, University of Virginia, 345 Crispell Dr. MR-6, Charlottesville, VA 22908, USA. [3] Robert M. Berne Cardiovascular Research Center, University of Virginia, 415 Lane Rd, Charlottesville, VA 22908, USA. [4] Present address: Department of Cell Biology, Harvard Medical School, Boston, MA, USA. ✉email: bdesai@virginia.edu

In a healthy human body, billions of cells die every day[1], and many of these are cleared by macrophages, the quintessential phagocytes resident in most tissues. *Efferocytosis* encompasses the cellular processes through which apoptotic cells are recognized, engulfed, and digested by other neighboring cells or the "professional" phagocytes that survey the tissue for sterile or pathogenic insults. The efficient removal of cellular corpses by the innate immune system shapes tissue development, wound repair, host defense, and organ homeostasis[2]. Uncovering the cellular logic, mechanisms, and molecules that govern efferocytosis is therefore an important area of basic research with meaningful, if not imminent, clinical implications.

In healthy tissue, macrophages sense, recognize, and engulf apoptotic cells in an immunologically silent manner—this promotes regenerative processes without the inflammatory and self-destructive activation of the immune system[3,4]. Macrophages employ a variety of cell surface receptors to recognize and engulf non-opsonized apoptotic cells. The mechanisms that control the early stages of engulfment and formation of a nascent phagosome have been well-characterized[5]. However, the machinery and signals regulating *phagosome maturation*, the intracellular process that "digests" an engulfed apoptotic cell without inflammatory fallout, remain substantially more enigmatic[6]. Therefore, defining the molecular components and signaling mechanisms of *"efferophagosome"* maturation is crucial for a clear understanding of tissue homeostasis, inflammation, and wound repair.

The nascent phagosome is not intrinsically destructive but it undergoes an extensive transformation during phagosome maturation[7]. The phagosome sequentially transitions from an 'early' to 'late' stage through the remodeling of its lipid, protein, and ionic constituents, culminating with the fusion of the phagosome with lysosomes. Aided by cytoskeletal rearrangements and vesicular fusion, the newly recruited phagosomal proteome promotes an increasingly acidic vacuolar compartment through the vacuolar ATPase (V-ATPase) complex activity, which pumps protons ($H^+$) into the phagosome via ATP hydrolysis[8]. Notably, injection of $H^+$ has a hyperpolarizing effect on the phagosome membrane, and the efficiency of V-ATPase pump must be sustained by a countercurrent of cations (e.g., $Na^+$, $Ca^{2+}$) from the phagosome to the cytosol[9]. The components and signals that regulate phagosome maturation remain poorly defined, but there is considerable evidence suggesting that $Ca^{2+}$-signaling plays a significant role in membrane fusion events involved in phagosome maturation[10,11]. However, the ion channels that modulate the crucial $Ca^{2+}$ flux and electrical activity during phagosome maturation remain poorly defined.

$Ca^{2+}$-signaling promotes the activity of key molecular actuators during phagocytosis, including certain phospholipases (involved in membrane remodeling), gelsolins (necessary for cytoskeletal rearrangements), and synaptotagmins (crucial for membrane fusion). Elevations in cytosolic $Ca^{2+}$ have long been observed to occur during phagocytosis[12–15], but the underlying molecular machinery has not been defined. Furthermore, the role of $Ca^{2+}$-signaling in phagocytosis has been studied primarily in Fc-receptor-mediated phagocytosis of opsonized cargo. In contrast to Fc-receptor mediated phagocytosis, it is not known whether $Ca^{2+}$-elevations occur during mammalian efferocytosis and whether $Ca^{2+}$-signaling is required for efferocytosis. Genetic studies in *C. elegans*[16,17] and *D. melanogaster*[18,19] suggest an important but undefined role for $Ca^{2+}$ signaling in efferocytosis. Even if efferocytosis is conjectured to require $Ca^{2+}$-signaling, the specific ion channels, the molecular conduits that mediate these spatiotemporal $Ca^{2+}$-signals, have not been identified. Since ion channels are appealing molecular targets for pharmacological intervention, the identification of ion channels that regulate efferocytosis in general and phagosome maturation in particular

will advance this field toward therapies involving immunomodulation and regenerative medicine.

In this study, we demonstrate that $Ca^{2+}$-signaling is necessary for the subsequent phagosome maturation of non-opsonized apoptotic cells—as defined by phagosome acidification. Since macrophages express a variety of $Ca^{2+}$ permeable channels that could potentially drive $Ca^{2+}$-signaling, we developed an siRNA screen to identify the pertinent ion channel(s). Through this screen, we identified the ion channel TRPM7 as a vital component of phagosome acidification. TRPM7 is comprised of a $Ca^{2+}$-conducting channel (non-selective, also conducts $Na^+$, $Zn^{2+}$, and $Mg^{2+}$) and a serine-threonine kinase domain. TRPM7 also regulates LPS-stimulated CD14/TLR4 signaling[20], positioning the channel at the crossroads of inflammation and tissue homeostasis. To define the function of TRPM7 in phagosome maturation, we used mouse lines that express genetically encoded $Ca^{2+}$ indicator GCaMP6s in myeloid cells and measured the $Ca^{2+}$-elevations associated with efferocytosis in *Trpm7*$^{+/+}$ and *Trpm7*$^{-/-}$ macrophages. During efferocytosis, TRPM7-dependent $Ca^{2+}$-elevations are localized proximal to the phagosome and peak prior to phagosome acidification. We also show that TRPM7 channel is activated by low pH, demonstrating that TRPM7 mediates a cationic current necessary for sustained acidification of the phagosome. Our study has identified a TRPM7-mediated $Ca^{2+}$-signaling module necessary for phagosome maturation during apoptotic cell clearance.

## Results

**Rapid, localized cytosolic $Ca^{2+}$ elevations are required for phagosome acidification during efferocytosis.** To study efferocytosis, we used a flow cytometry-based assay to measure engulfment and acidification of apoptotic cell cargo stained with a fluorescent dye sensitive to pH[21] (Fig. 1a). UV-induced apoptosis yielded an "early apoptotic" (Annexin V+, 7AAD−) cellular population (Supplementary Fig. 1a, b). Co-staining with fluorescent dyes which exhibit an orthogonal response to pH allowed simultaneous measurement of cargo engulfment (via CellTrace Violet) and acidification (CypHer5E) during phagocytosis. CypHer5E-stained cells display a linear, inversely proportional increase in fluorescence with pH (lowering pH increases CypHer5E fluorescence), while CellTrace Violet fluorescence is stable across a broad range of pH, and only diminished at very acidic pH (pH < 4.5) but still readily identifiable via flow cytometry. (Supplementary Fig. 1c, d). Using flow cytometry, we validated this method by verifying that Bafilomycin A1 (inhibitor of V-ATPase) prevented phagosome acidification without impairing the initial engulfment or binding by bone-marrow-derived macrophages (BMDMs) (Supplementary Fig. 1e, f). As an additional experimental control, we also pretreated BMDMs with Cytochalasin D (inhibitor of actin polymerization) to prevent engulfment of apoptotic cells but not cargo binding to the cell surface. Together, these reagents established a robust approach to quantify engulfment and phagosome maturation simultaneously.

We first tested the role of $Ca^{2+}$ signaling in macrophage efferocytosis. BMDMs were loaded with either BAPTA-AM or EGTA-AM $Ca^{2+}$ chelators prior to incubation with labeled apoptotic cells. Although BAPTA and EGTA have similar affinities for $Ca^{2+}$, they exhibit very different kinetics of $Ca^{2+}$-binding[22]. BAPTA, a 'fast' $Ca^{2+}$ chelator, prevents cytosolic $Ca^{2+}$-elevations so efficiently that even local $Ca^{2+}$-elevations at the mouth of the $Ca^{2+}$-conducting channel are prevented. In contrast, the slow chelation of $Ca^{2+}$ by EGTA prevents sustained global $Ca^{2+}$-elevations but allows local $Ca^{2+}$-elevations or $Ca^{2+}$ 'puffs'[23]. To test these chelators, we stimulated BMDMs with ATP, an important 'find-me' signal released by apoptotic

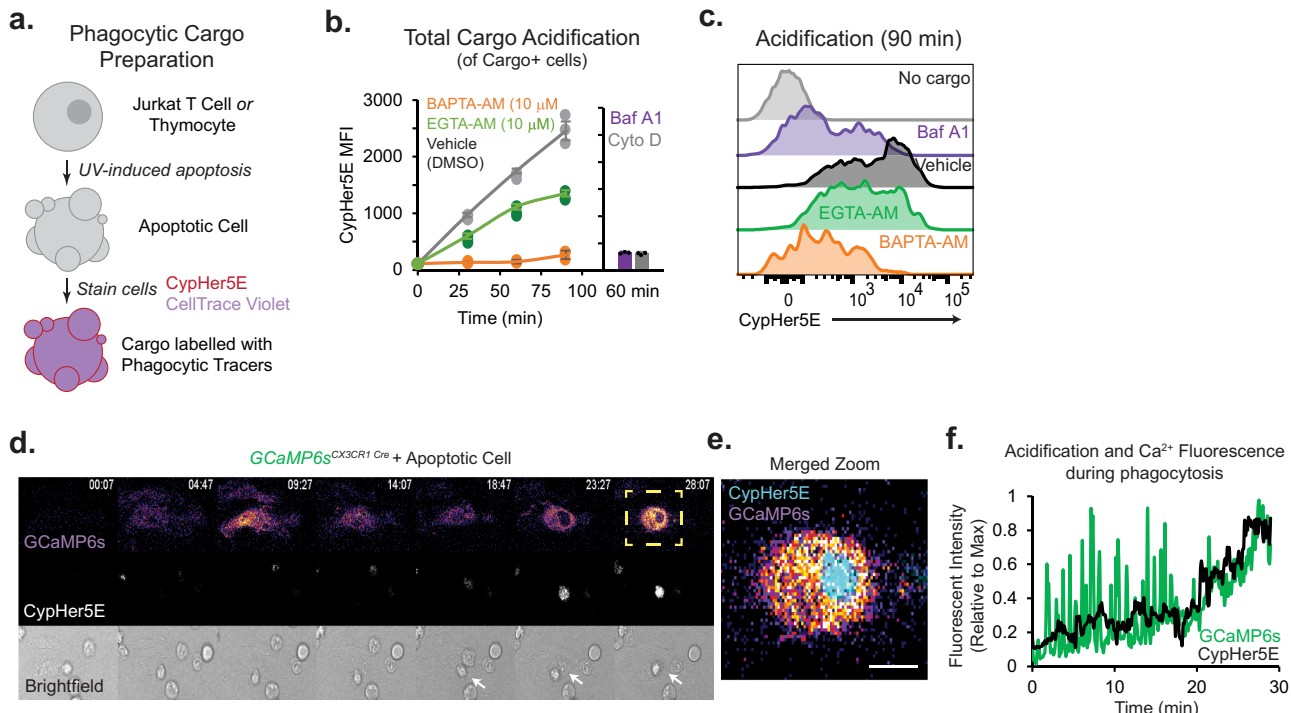

**Fig. 1 Ca²⁺ signaling requirements of phagosome acidification. a** Preparation of apoptotic cell cargo is schematized. UV-irradiated apoptotic cells are stained with pH-insensitive (CellTrace Violet) and pH-sensitive (CypHer5E) dyes to measure cargo association and acidification, respectively. Apoptotic cells are then offered to macrophages by centrifugation, and unengulfed cells are removed by trypsinization and washes in Ca²⁺-free saline. **b** Flow cytometry-based measurement of acidification of phagosomes after engulfment of apoptotic cell cargo by BMDMs loaded with vehicle (DMSO), BAPTA-AM or EGTA-AM (both 10 µM, 30 min). Engulfing CD11b + BMDMs containing labeled cargo (CellTrace Violet) were gated and analyzed for acidification (CypHer5E fluorescence). Bafilomycin A1 (BafA1; 500 nM) and Cytochalasin D (Cyto D; 1 µM) treated cells were used as negative controls. Data points indicate independent samples from different animals; bar charts represent mean value (Error bars = SD). $p < 0.001$ for all statistical comparisons between treatment conditions by two-way ANOVA (by Time and Treatment [$F_{(4,12)}$ 30.79]) with Bonferroni's correction. **c** Representative histograms showing phagosome acidification after 90 min of phagocytosis. Data are representative of biological triplicate samples described in **b**. **d** [Ca²⁺]ᵢ dynamics in GCaMP6s-expressing BMDMs during phagocytosis of apoptotic Jurkat cells. GCaMP6s (top image; Fire LUT), CypHer5E fluorescence (middle; gray), and brightfield (bottom image; gray) were acquired using wide-field microscopy (1 frame/7 s). Region of interest (ROI) shown in yellow box is magnified in panel **e**. White arrows indicate engulfed cell. See also Supplementary Movie 1. **e** ROI outlined in panel **d**. is shown as a merged image of GCaMP6s (Fire LUT) and CypHer5E (Cyan) fluorescence, 28 min after the addition of apoptotic cells to BMDMs. Scale bar is 10 µm. **f** Quantification GCaMP6s (green trace) and CypHer5E (black trace) fluorescence intensity over time. Measurements are from data depicted in panel **d** over time. Source data are provided as a Source Data file, and statistical testing is described in "Statistics and Reproducibility".

cells[24,25]. Consistent with their reported kinetics, we observed that BAPTA-AM (10 µM) abolished ATP-induced Ca²⁺ elevations completely whereas EGTA-AM (10 µM) only attenuated and delayed ATP-induced Ca²⁺ responses in Fura-2-loaded BMDMs (Supplementary Fig. 1g). During efferocytosis, we found that BAPTA-AM-loaded BMDMs fail to acidify the phagosomes (90.3% decrease compared to vehicle-treated BMDMS), while EGTA-AM-loaded macrophages show a relatively modest effect in acidification (only 48.5% decrease when compared to vehicle-treated BMDMs) (Fig. 1b, c). In BMDMs that associated with cargo, neither chelator decreased engulfment (Supplementary Fig. 1h), and in fact, BAPTA-AM-loaded BMDMs appear to display increased cargo aggregation—likely because the BMDMs accumulate the labeled cargo, or fail to digest their cargo when phagosome maturation is blocked completely. To test if these Ca²⁺ chelators influenced basal lysosome function, thereby influencing efferophagosome acidification, we monitored the accumulation of an acidotropic dye or degradation of DQ-green-BSA, which fluoresces upon hydrolysis by lysosomal proteases[26], in BMDMs loaded with the Ca²⁺ chelators. We did not observe a significant effect of BAPTA-AM or EGTA-AM loading on lysosomal acidity or proteolytic activity (Supplementary Fig. 1i, j). These observations indicate that 'fast' chelation of Ca²⁺ by

BAPTA-AM, which chelates all cytosolic Ca²⁺ changes, completely inhibits phagosome acidification. In contrast, EGTA-AM, which cannot chelate rapid, localized Ca²⁺ 'puffs' efficiently, has a relatively minor effect on phagosome acidification. However, we acknowledge that additional, yet undefined, differences in BAPTA-AM and EGTA-AM, such as in their relative cytosolic accumulation, could also account for the differences.

Although Ca²⁺-elevations have been measured during Fc-receptor mediated phagocytosis, the spatiotemporal dynamics of Ca²⁺-elevations during efferocytosis (non-opsonized target) are not known. Macrophages pump out small molecule Ca²⁺ indicator dyes such as Fluo-4-AM or Fura-2-AM very efficiently and thus require the concomitant use of broad-spectrum pump blockers such as Probenecid. However, because of non-specific effects on pumps, ion channels, and transporters[27–29], such blockers of dye efflux are unsuitable for the study of phagosome maturation. Thus, to carry out these studies, we generated mice that express GCaMP6s in myeloid cells (GCaMP6sCX3CR1-Cre; herein, "WT GCaMP6s"). Using live-cell fluorescence microscopy, we monitored GCaMP6s-expressing BMDMs during phagocytosis of apoptotic cells (Supplementary Movie 1). These BMDMs show clear cytosolic Ca²⁺ oscillations (~1 every 49 s) upon 'sensing' apoptotic cargo and during the initial stages of

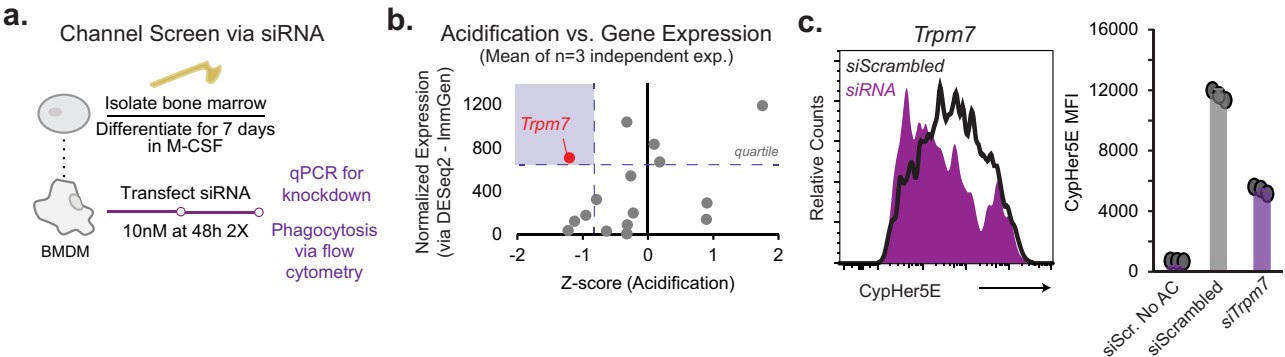

**Fig. 2 Identification of the ion channel TRPM7 as a regulator of phagosome acidification. a** Schematic showing the siRNA knockdown of candidate calcium channels in BMDMs. **b** Scatter plot showing the relationship between macrophage gene expression and relative effect on phagosomal acidification, depicted as mean Z-scores ($n = 3$) from siRNA channel screen; target genes are depicted by single point; derivation of Z-scores is described in *Methods*. Gene expression value is average expression level across four macrophage subsets (ImmGen). Gray box denotes quartile for highest gene expression and the most negative effect on the z-score of acidification. **c** Left panel shows overlaid histograms of acidification (CypHer5E fluorescence) in the Trpm7 knockdown (*siTrpm7*, purple) and control (*siScrambled*, black) cells. Right panel shows Mean CypHer5E mean fluorescent intensity (MFI) of CypHer5E. The bar graphs depict mean values and error bars reflect SD ($n = 3$ biological replicates). Data representative of $n = 3$ independent experiments. See also Supplementary Figs. 1 and 2. Source data are provided as a Source Data file, and statistical testing is described in "Statistics and Reproducibility".

engulfment (Fig. 1d). However, as the cargo is steadily acidified, the macrophages display a sustained rise in cytosolic $Ca^{2+}$ that correlates with phagosome acidification (Fig. 1e, f). The dichotomy of BAPTA-AM and EGTA-AM influence on cargo acidification suggests that the maturation-associated $Ca^{2+}$-signals are local and proximal to the phagosome. Together, these results demonstrate that cytosolic $Ca^{2+}$-elevations occur during efferocytosis and $Ca^{2+}$-signaling is necessary for phagosome maturation.

**Identification of $Ca^{2+}$ channel regulators of phagosome acidification.** Since Phospholipase C is activated during efferocytosis[1,30], cytosolic $Ca^{2+}$ elevations may involve store-operated $Ca^{2+}$ entry (SOCE) through the Orai channels. However, recent studies have reported that macrophages deficient in Stim1 and Stim2, crucial components of SOCE machinery, have no defects in phagocytosis[31] and furthermore, those experiments did not use apoptotic cell cargo. To identify candidate channels that may facilitate efferophagosome maturation-associated $Ca^{2+}$ signaling, we used publicly available (Immunological Genome Project[32]) to identify $Ca^{2+}$-conducting ion channels that are expressed unambiguously in macrophages (Supplementary Fig. 2a). After expression validation by qPCR, we devised an siRNA-based knockdown protocol to effectively deplete (~80% reduction) the 14 candidate ion channel mRNA in primary BMDMs (Fig. 2a and Supplementary Fig. 2b). We also tested a siRNA double-knockdown of Stim1 and Stim2, which are required for SOCE. Using flow cytometry, we measured the acidification of engulfed apoptotic Jurkat cells after 90 min of phagocytosis by siRNA-treated BMDMs for candidate ion channels (Supplementary Fig. 2c).

To identify regulatory candidates, we converted acidification (CypHer5E MFI) to z-scores relative to control scrambled siRNA (siScrambled)-treated BMDMs. The z-scores were then plotted in relation to the average gene expression level across four macrophage subsets (ImmGen Project) (Fig. 2b). Consistent with the previous observations[31], double-knockdown of Stim1 and Stim2, did not substantially inhibit phagosome acidification. Through this analysis, we identified ion channels that regulate phagosome acidification positively and negatively and are the subject of independent ongoing studies. Strikingly, when considering ion channel candidates in the top quartile for both gene expression and z-score, only Trpm7 emerged as a candidate

crucial for phagosome acidification (marked in red, Fig. 2b), as knockdown of Trpm7 results in a 53% decrease in acidification relative to siScrambled-treated BMDMs (Fig. 2c). As expected, Bafilomycin A1, BAPTA-AM, and Cytochalasin D, inhibited cargo acidification (at least 60% decrease relative to vehicle-treated BMDMs) (Supplementary Fig. 2d). Interestingly, the $Ca^{2+}$ ionophore Ionomycin inhibited phagosome acidification. This suggests that a sustained, global influx of $Ca^{2+}$ is detrimental to phagosome maturation; alternatively, Ionomycin may provide a conduit for phagosomal $H^+$ efflux[33] to inhibit maturation. In summary, the siRNA screen identified TRPM7 as a major candidate for a channel that regulates phagosomal acidification in BMDMs.

**TRPM7 regulates phagosomal acidification during macrophage efferocytosis.** To test the hypothesis that TRPM7 regulates phagosome acidification, we derived BMDMs from mice wherein Trpm7 is deleted in myeloid cells [Trpm7$^{fl/fl}$ (LysM Cre)]—generated previously by our group[20]. Here, these Trpm7-deficient BMDMs ("KO BMDMs") are compared to BMDMs derived from Trpm7$^{fl/fl}$ mice ("WT BMDMs"). We used LysoTracker dye to fluorescently label the acidic compartments during efferocytosis—this dye labels acidic cellular compartments, including lysosomes, but is especially useful for the detection of the highly acidic and large phagolysosomes formed during efferocytosis. Labeled apoptotic cells were added to LysoTracker-stained BMDMs for 90 min and fixed prior to confocal microscopy. No gross differences in lysosomal staining were observed between WT and KO BMDMs, prior to efferocytosis (Fig. 3a). However, after 90 min of efferocytosis, the KO BMDMs show a striking defect in the formation of large, highly acidic phagolysosomes when compared to WT BMDMs (Fig. 3a, b). We also used flow cytometry to measure engulfment and phagosome acidification in a large population of cells. Although Trpm7-deficient BMDMs show only a modest deficit in engulfment of apoptotic cells, they exhibit a striking decrease in efferophagosome acidification compared to WT BMDMs (Fig. 3c, d) and the acidification defect was comparable to Bafilomycin A1-treated cells. Moreover, we observed no significant difference in basal lysosomal pH between WT and KO BMDMs, based on the pH-dependent accumulation of an acidotropic dye (LysoSensor) (Supplementary Fig. 3a). Together, these results clearly indicate TRPM7 regulates phagosome acidification during phagocytosis. Next, we tested whether deletion of

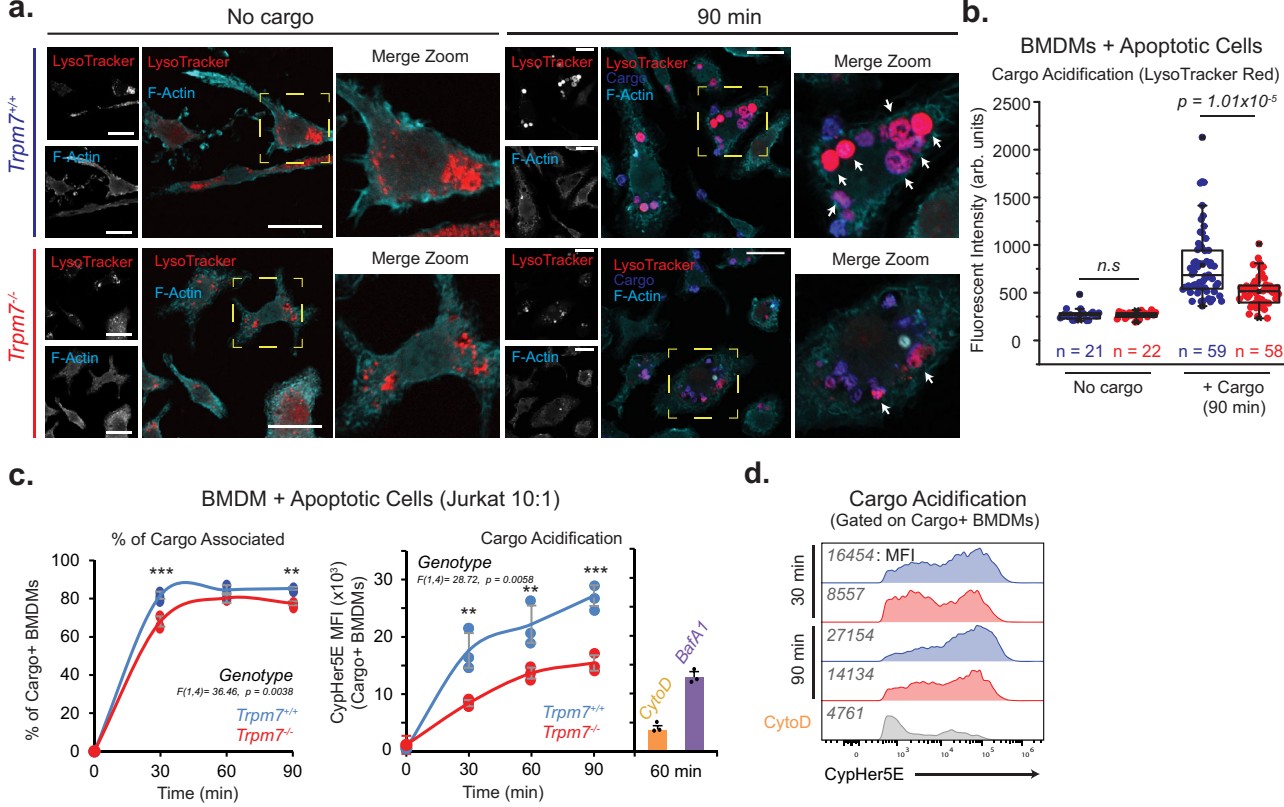

**Fig. 3 TRPM7 is required for acidification of apoptotic cells during efferocytosis. a** Confocal immunofluorescence microscopy of BMDMs with or without addition of apoptotic cells as cargo for phagocytosis. Apoptotic cells were labelled with CellTrace Violet (Cargo-blue) and live BMDMs were stained with LysoTracker (red) prior to PFA-fixation and then phalloidin stained (cyan) after fixation. Single-channel images of LysoTracker and F-actin (phalloidin) are shown with merged pseudocolored image. Yellow dotted box indicates zoomed ROI of adjacent image and arrows indicate phagolysosomes. Single optical sections (0.45 μm) are shown. Scale bar = 20 μm. Representative images are from three independent experiments. **b** Quantification of LysoTracker fluorescent intensity, as shown in panel **a**. Data points represent intensity measurement for single phagocyte; n values for each condition are included in the figure. The parameters included in box charts are described in the methods. **c** Flow cytometry-based measurement of phagocytosis of apoptotic Jurkat cell cargo by BMDMs (J:B = 10:1) over time. Cargo were stained with CellTrace Violet and CypHer5E prior to adding to BMDMs at level indicated in figure. Engulfing BMDMs were gated for anti-mCD45 antibody staining prior to the measurement of CellTrace Violet and CypHer5E staining. Control samples with Cytochalasin D (1 μM) and Bafilomycin A1 (500 nM) were pretreated 15 min prior to the addition of AC cargo. Data points represent n = 3 independent samples, and the data are representative of three independent experiments. Error bars represent SD centered on mean. **d** Representative histograms of data in panel **c**. Mean fluorescent intensity values are as indicated on figure panel. Representative of n = 3 independent samples. Source data are provided as a Source Data file, and statistical testing is described in "Statistics and Reproducibility".

*Trpm7* from myeloid cells impairs the degradation of apoptotic cells in vivo.

**Myeloid cells require TRPM7 for phagosome maturation of apoptotic cells in an in vivo model of efferocytosis.** To measure the phagocytic and degradative capacity of myeloid cells in the peritoneal cavity, we administered apoptotic cells intraperitoneally to *Trpm7*$^{fl/fl}$ *(LysM Cre)* and *Trpm7*$^{fl/fl}$ mice as a model of efferocytosis by peritoneal macrophages and myeloid cells in situ. This model does not rely on exogenous adjuvants to recruit or activate peritoneal cells, allowing measurement of homeostatic phagocytic activity in *Trpm7*-deficient myeloid phagocytes. As apoptotic cargo, we utilized Jurkat cells that stably express GFP and labeled with CypHer5E ("Jurkat-GFP"). GFP fluorescence is expected to decrease in response to acidic pH[34] and due to protein degradation, thereby permitting GFP fluorescence to report phagosome maturation as a loss of signal. As expected, when apoptotic Jurkat-GFP are engulfed by cultured BMDMs, their GFP fluorescence diminished over time of engulfment with a concomitant rise in the acidification, as measured by CypHer5E MFI, validating that loss of GFP fluorescence

can be used as an indicator of cargo digestion by macrophages (Supplementary Fig. 3b). When cultured alone after UV-induced apoptosis, Jurkat-GFP cells retain GFP fluorescence for 90+ min and show no spontaneous increase in CypHer5E fluorescence. Using Jurkat-GFP cells, we designed an in vivo readout of apoptotic cell clearance (schematized in Fig. 4a) by injecting co-stained Jurkat-GFP cells in the mouse peritoneum that permits simultaneous measurement of engulfment (CellTrace Violet), acidification (CypHer5E), and cargo degradation (GFP), albeit with the unavoidable caveat that the loss of GFP fluorescence reflects a combination of phagosome acidification and protein degradation.

We first characterized the proportion of peritoneal macrophages in TRPM7 WT and KO mice, as differences in this population could account for experimental variability. There were no significant differences in the proportion or number of CD11b+ peritoneal cells or macrophages (F4/80+, CD11c-) between *Trpm7*$^{fl/fl}$ and *Trpm7*$^{fl/fl}$*LysM Cre* mice (Supplementary Fig. 3c, d). Approximately 50% of peritoneal cells were myeloid cells and 95% of these cells were peritoneal macrophages. These results indicate that any differences in phagocytosis are due to functional differences between WT and *Trpm7*$^{-/-}$ myeloid

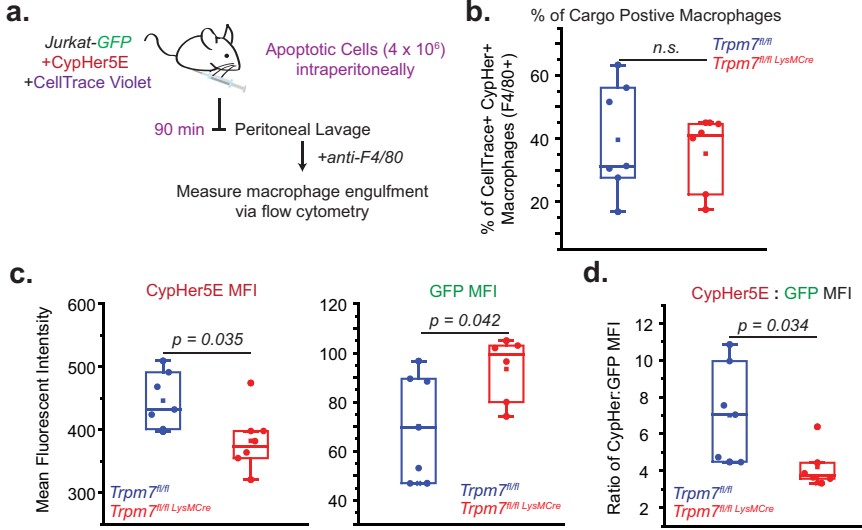

**Fig. 4 Degradation of apoptotic cell cargo by peritoneal macrophages requires TRPM7. a** The experimental design to measure apoptotic cell clearance in vivo is schematized. **b** Quantification of engulfing peritoneal macrophages (F4/80+) in *Trpm7*<sup>fl/fl</sup> (*n* = 8) and *Trpm7*<sup>fl/fl</sup> *LysM Cre* (*n* = 7) mice during in vivo clearance of apoptotic Jurkat-GFP cells. **c** MFI of CypHer5E, which measures cargo acidification (left) and GFP, which is stably expressed in Jurkat-GFP cells (right). See also Supplemental Fig. 3 for gating strategy. **d** Quantification of cargo degradation by WT and KO peritoneal macrophages in vivo as depicted by the ratio of CypHer5E MFI to GFP MFI in Cargo+ peritoneal macrophages. Dots represent independent samples from individual mice in *n* = 3 independent experiments. Box charts display sample distribution and are described in Methods. Source data are provided as a Source Data file, and statistical testing is described in "Statistics and Reproducibility".

cells—not differences in numbers of myeloid cells. Apoptotic Jurkat-GFP cells (4 × 10⁶ total cells), labeled with CypHer5E and CellTrace Violet, were injected into the peritoneal cavity of mice and after 90 min, peritoneal cells were collected by lavage and analyzed by flow cytometry. The gating strategy used for flow cytometric analysis is shown in Supplementary Fig. 3e. There were no differences in the percentage of cargo-associated phagocytes between *Trpm7*<sup>fl/fl</sup> and *Trpm7*<sup>fl/fl</sup> *LysM Cre* mice (Fig. 4b) reflecting similar engulfment capacity of apoptotic cells. However, in the peritoneum of *Trpm7*<sup>fl/fl</sup> *LysM Cre* mice, the *Trpm7*<sup>−/−</sup> macrophages failed to acidify their phagosomes (low CypHer5E MFI) and degrade their cargo (GFP MFI remained high) (Fig. 4c). This is especially striking when visualized as a ratio of CypHer5E and GFP MFI (Fig. 4d). These in vivo results substantiate the hypothesis that TRPM7 is required for phagosome maturation. Although the data suggest that TRPM7 may also promote phagolysosomal assembly, since GFP fluorescence is also sensitive to pH, these results do not allow us to conclude definitively that TRPM7 regulates the formation of phagolysosome. To test that specific hypothesis, we used a more direct assay of phagolysosomal proteolytic activity.

**TRPM7 regulates phagosome proteolysis during cargo maturation.** The fusion of lysosomes with the phagosome is the penultimate stage of cargo degradation. Although the acquisition of lysosomal markers on the phagosome may indicate close proximity of the organelles, direct measurement of phagosomal proteolytic activity is the optimal readout of the phagosome-lysosome fusion[35]. To test the hypothesis that TRPM7 regulates phagolysosome formation, we utilized an assay that measures the proteolytic activity of the phagosome during phagocytosis. Fluorescent, polystyrene beads were conjugated to fluorescent DQ-Green-BSA, a BODIPY dye conjugate[36]. The fluorescence of DQ-Green is quenched when BSA is heavily labeled with the dye. Acidification does not increase DQ-Green fluorescence, but upon proteolysis of DQ Green-conjugated BSA by proteases, the quenching is relieved producing green fluorescence. This contrasts with GFP + Jurkats, where the loss of fluorescence could be

accounted for by the sensitivity of GFP to pH or proteolysis. Thus, DQ-Green-BSA beads report proteolytic activity in the phagosome by a gain of DQ-Green fluorescence signal (Fig. 5a), which is presumably derived from lysosomal proteases. WT and KO BMDMs were incubated with DQ-Green- BSA-labelled beads, and the fluorescence was measured by flow cytometry at varying time points. As expected, Bafilomycin pretreatment had a profound decrease in phagolysosomal proteolytic activity (65% reduction relative to vehicle-treated WT BMDMs) (Fig. 5b). Both WT and KO BMDMs showed a similar capacity for cargo uptake (both >90% cargo+ by 30 min) and amount of cargo engulfed (Fig. 5c). However, phagosomal proteolysis, as shown by DQ-Green fluorescence, was decreased significantly in KO BMDMs at 30, 60, and 90 min (47, 33.5, and 24.3% decrease, respectively, relative to WT) after incubation with the beads (Fig. 5d). Although statistically significant, we do not consider the 3% average difference in the engulfment of apoptotic cells to be a salient defect because it likely arises as an effect that is secondary to the primary defect in phagosome maturation. For instance, it is likely that disruption in phagosome maturation slows the flux of the phagocytic pathway and results in a lower rate of engulfment. These results suggest that *Trpm7*-deficient BMDMs have decreased phagosomal proteolytic activity, likely due to diminished phagolysosome fusion. Collectively, these results clearly demonstrate that TRPM7 is required for phagosome maturation and provide evidence that optimal phagosomal proteolytic activity, likely regulated by phagolysosome formation, depends on TRPM7 activity during efferocytosis.

**TRPM7 localizes proximal to the nascent phagosome.** The subcellular localization of TRPM7 remains poorly understood. TRPM7 is present on the macrophage plasma membrane[20], it has also been reported in the membrane of small intracellular vesicles in other cell types[37,38]. To facilitate efferocytosis, TRPM7 may be incorporated into the nascent phagosome from the cell membrane, recruited from vesicular pools proximal to the phagosome, or support a cellular activity distinct from the phagosome. Like many ion channels, immunolocalization of native

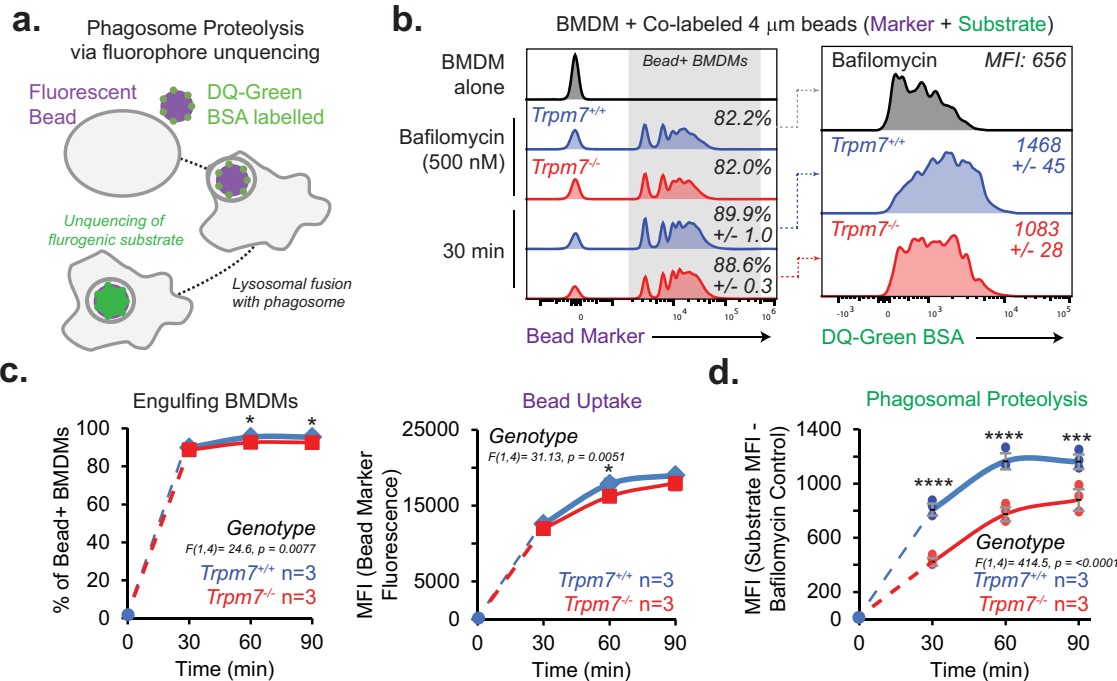

**Fig. 5 TRPM7 regulates phagosome proteolysis in macrophages. a** Schematic of phagosomal proteolysis assay. Fluorescent beads were conjugated to DQ green-BSA and then offered to BMDMs as cargo for phagocytosis. In bead-containing macrophages, lysosomal proteolytic activity was measured, as reflected by increased DQ-Green BSA fluorescence due to proteolytic unquenching of the fluorophore. **b** Flow cytometry-based measurement of proteolytic activity during phagocytosis of latex beads. DQ-green BSA-labelled fluorescent beads (4 μm) were incubated with WT and KO BMDMs for indicated time points. Left: Representative histograms of Bead+ BMDMs. Right: Measurement of DQ-green MFI in Bead+ BMDMs. Data representative of two independent experiments. **c** Quantification of bead uptake by $Trpm7^{+/+}$ and $Trpm7^{-/-}$ BMDMs. Left: Percentage of Bead+ BMDMs over time. Right: Quantification of gross bead uptake measured by Bead MFI of Bead+ BMDMs. Data points are mean of $n = 3$ independent samples. Error bars = SEM. **d** Quantification of the phagosomal proteolysis as shown in panel **c**. DQ-green BSA MFI is shown as change from Bafilomycin A1 pretreated BMDMs (negative control) and gated on Bead+ BMDMs. Individual sample data points are plotted. Error bars = SEM. Source data are provided as a Source Data file, and statistical testing is described in "Statistics and Reproducibility".

TRPM7 has posed a major technical hurdle and this is further exacerbated by the lack of high-quality antibodies targeting TRPM7. To circumvent these challenges, we transfected FLAG-TRPM7[39] into RAW 264.7 cells, a murine macrophage cell line, and examined its subcellular location during efferocytosis using immunofluorescence-labeling and ImageStream flow cytometry, which simultaneously acquires microscopy images of cells as they pass through the flow cell. This method enables the measurement of thousands of cells and quantitative colocalization measurements of fluorescent labels, termed a "similarity score"—in this case, a similarity score between the fluorescence signals emitted by immunostained TRPM7 and fluorescent cargo (Fig. 6a). Ectopic expression of GFP was used as a negative control, as it should be expressed throughout the cytosol and not specifically associated with the phagosome. As expected, CellTrace-labeled apoptotic cells engulfed by GFP-expressing RAW 264.7 cells displayed low colocalization with GFP (−0.15 mean similarity score) (Fig. 6b and Supplementary Fig. 4a-c). In contrast, FLAG-TRPM7 exhibited significant colocalization with the engulfed cargo after 60 min of phagocytosis (0.35 mean similarity score). Interestingly, neither treatment with Cytochalasin D nor Bafilomycin A1 significantly altered cargo colocalization with TRPM7, suggesting that TRPM7 is localized near the contact site of the cargo itself. Overexpression of TRPM7 resulted in an increase in association with apoptotic cell cargo and a significant increase in cargo acidification, compared to GFP alone (Supplementary Fig. 4a).

To resolve the subcellular localization of TRPM7 during phagocytosis, we immunostained FLAG-TRPM7 in transfected LR73 cells—large phagocytic cells with the ability to engulf apoptotic cells that are ideal for confocal microscopy[21,40]. TRPM7 was readily observed proximal to the phagosome during phagocytosis (Fig. 6c and Supplementary Figs. 4d, e, 5a, File 1, and Movie 2). As reported previously, we detected TRPM7 at the plasma membrane and vesicular TRPM7 is found distributed in the cytoplasm of these cells. Co-immunostaining with a marker for the endoplasmic reticulum (ER) indicated that FLAG-TRPM7 localized to a cellular sub-domain proximal to the phagosome and distinct from the ER (Fig. 6c and Supplementary Fig. 4e), indicating that the subcellular localization pattern was not an overexpression artifact resulting from unfolded TRPM7. Together, these results indicate that TRPM7 is recruited to the nascent phagosome, but at this point it is not known whether it is recruited from the plasma membrane or through the fusion of the TRPM7-containing vesicles.

**TRPM7 channel activity is essential for phagosome maturation.** The activity of TRPM7 may be regulated by changes in the membrane phospholipids[41] or the initial acidification of the phagosome, which could activate TRPM7 channel[42]. To test whether FTY720, a TRPM7 channel blocker[43], could also inhibit phagosome maturation, BMDMs were pretreated with FTY720 (5 μM) 15 min prior to the addition of apoptotic cell cargo, and the inhibitor remained in the media for the duration of the assay. There was a modest decrease in engulfment at 30 min in FTY720-treated BMDMs compared to media alone, but no difference was observed at later time points (Fig. 7a and Supplementary Fig. 5b, c). However, FTY720 decreased phagosomal acidification

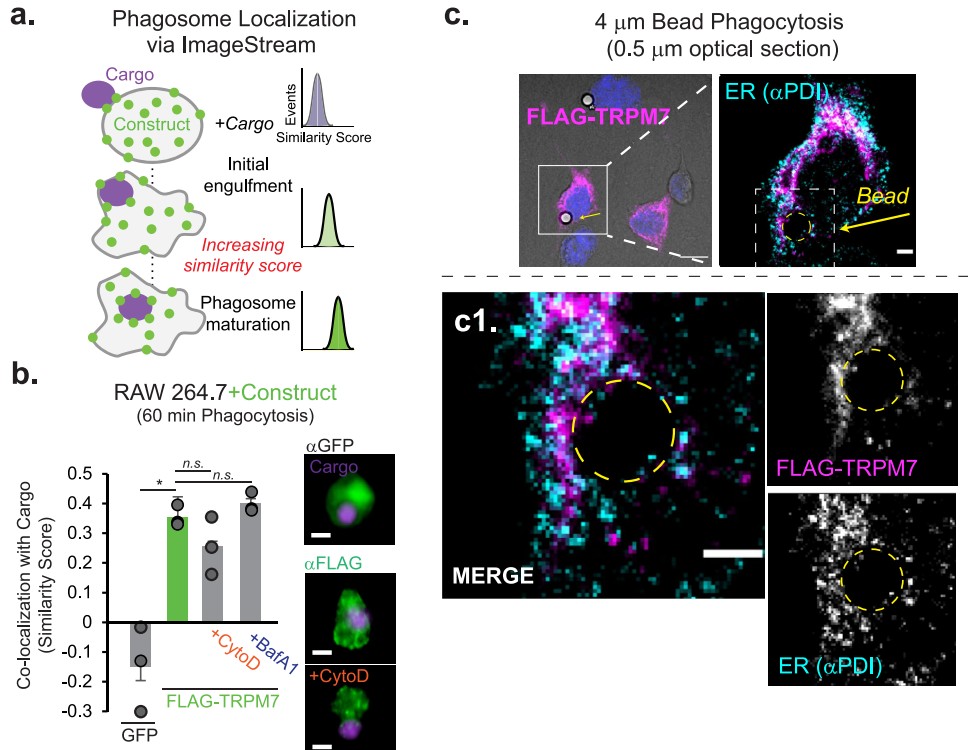

**Fig. 6 TRPM7 localizes proximal to the nascent phagosome. a** ImageStream-based analysis of protein localization in RAW 264.7 cells during phagocytosis of apoptotic cells is schematized. Similarity score measures colocalization of expression construct with CellTrace Violet-labelled cargo, and the score increases with increased colocalization. **b** ImageStream-based measurement of TRPM7 localization with phagocytosed apoptotic cells in fixed RAW 264.7 cells. Either FLAG-TRPM7 or GFP was transfected 16 h prior to incubation with apoptotic cells. For controls, the cells were subjected to 15 min pretreatment with Cytochalasin D (Cyto D; 1 μM) or Bafilomycin A1 (BafA1; 500 nM). After phagocytosis, the RAW 264.7 cells were fixed and immunostained for FLAG immunoepitope. Fluorescence of GFP and anti-FLAG was measured via ImageStream flow cytometry. Bar charts represent mean of $n = 3$ independent samples and the individual data points are overlayed; error bars are SEM. Representative cells are shown at right. Representative results of two independent experiments are shown. *$p = 0.035$. See also Supplementary Fig. 4a–c. **c** Confocal immunofluorescence microscopy of FLAG-TRPM7-expressing LR73 phagocytes after addition of 4 μm beads for phagocytosis. Fixed cells were immunostained for FLAG (TRPM7-green) and fluorescent beads (red). Top left: Brightfield shown merged with FLAG (magenta) and nuclei (blue); scale bar = 10 μm. Top right: merged ROI of FLAG and ER marker (PDI). Bottom left [**c1**]: FLAG-TRPM7 and PDI light are shown with merged pseudocolored image; scale bar = 3 μm. Bottom right: Single-channel images shown at bottom left. White dash box indicates ROI, and yellow dashed circled shows bead-containing phagosome. Single optical sections (0.5 μm) are shown. Additional single-channel images shown in Supplementary Figs. 4d, e and 5a. Source data are provided as a Source Data file, and statistical testing is described in "Statistics and Reproducibility".

significantly, resulting in at least a 40% decrease in acidification across all time points (Fig. 7a). Although the role of the kinase activity of TRPM7 in phagosome maturation cannot be ruled out, the sensitivity of phagosome maturation to the TRPM7 channel blocker (FTY720) argues that the channel activity plays an important role in the regulation of phagosome maturation. At concentrations sufficient to activate sphingosine-1-phosphate receptors (10 nM) but not inhibit TRPM7 channel[43], FTY720 does not inhibit macrophage efferocytosis[44], indicating that FTY720 suppresses efferocytosis through inhibition of TRPM7 channel.

Activation of TRPM7 by PIP2 hydrolysis or another unknown mechanism may be key to triggering TRPM7-mediated $Ca^{2+}$-influx. Additionally, after the initial acidification, a sustained monovalent cationic current through TRPM7 may play an important role in the maintenance of phagosomal pH. The pH-dependent activation of a monovalent $I_{TRPM7}$ has been reported earlier[45] in 293 T cells using whole-cell configuration (WCC). WCC electrophysiology results in drastic dialysis of intracellular components and is highly disruptive to the cytoskeletal and intracellular structures that may be important for channel regulation. Alternatively, as illustrated in Fig. 7b, perforated-patch configuration (PPC) permits in situ recordings of cellular

channel activity without extensive disruption of intracellular signaling[41]. Instead of rupturing the plasma membrane to gain electrical access, PPC uses pore-forming compounds to induce small ion-permeable pores in the membrane patch at the recording pipette-electrode to facilitate the measurement of plasma membrane channel currents. Through PPC, we tested whether macrophage-resident TRPM7 is activated by low pH.

When BMDMs are recorded in PPC, $I_{TRPM7}$ is robustly activated by low pH (pH = 4.0–5.5), eliciting a characteristic outwardly rectifying $I_{TRPM7}$ (Fig. 7c and Supplementary Fig. 5d), which is readily blocked by a TRPM7 channel blocker (5 μM FTY720). A switch to pH 4.0 yields a two-fold and eight-fold increase in inward (at −100 mV) and outward (+100 mV) current densities, respectively. $I_{TRPM7}$ was induced almost immediately (<2 s) upon switch to pH 4.0. Next, we tested if TRPM7 can also be activated by pH 6.5 and pH 5.5. Using PCC, we recorded $I_{TRPM7}$ in WT and $Trpm7^{-/-}$ macrophages. These results clearly show that in WT cells, pH 5.5, but not pH 6.5, activates $I_{TRPM7}$ (Fig. 7d). Historically, isolation of $I_{TRPM7}$ has been substantiated by the demonstration that an application of 10 mM $Mg^{2+}$ blocks the isolated current[46]—this is the case here. More importantly, the veracity of the recorded $I_{TRPM7}$ is clearly evident by the fact that $Trpm7^{-/-}$ macrophages do not register

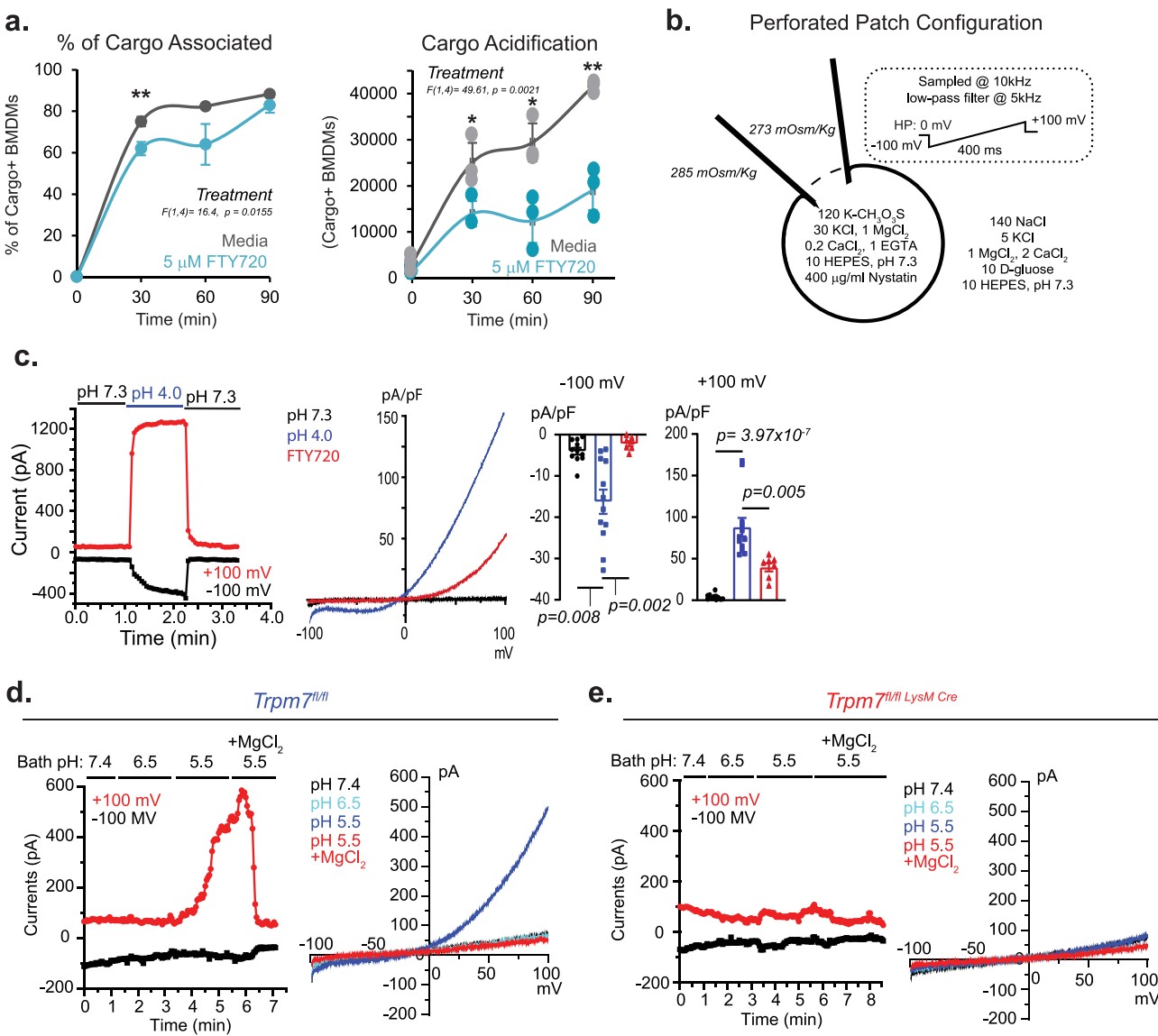

**Fig. 7 TRPM7 is activated by acidic pH in macrophages. a** Flow cytometry-based measurement of phagocytosis in BMDMs treated with media or FTY720 (5 μM) during phagocytosis of apoptotic Jurkat cells (J:B = 10:1). Left panel shows cargo association with BMDMs (CellTrace Violet+ CypHer5E+). Right panel shows acidification of engulfed cargo (Cypher5E MFI). Data points are mean of $n = 3$ independent samples; error bars are SD. Results are typical of two independent experiments. **b** Schematic of perforated-patch configuration used for electrophysiology. **c** Perforated-patch electrophysiology of $I_{TRPM7}$ in BMDMs in response to extracellular pH. Left: Time-current relationship of $I_{TRPM7}$ activation by bath pH as revealed in the perforated-patch recording. Middle: Representative I–V relationship of $I_{TRPM7}$ and block by FTY720 at pH 4.0. Right: Quantification of current densities at −100 mV (inward current) and +100 mV (outward current). Dots represent individual cell recordings with quantification under each condition shown. Error bars are SEM. **d** pH-dependent activation of $I_{TRPM7}$ in WT BMDMs using perforated-patch electrophysiology. Left: Time-current relationship of $I_{TRPM7}$ activation and block by 10 mM $MgCl_2$. Right: Representative I–V relationship of perforated-patch-clamp $I_{TRPM7}$ current in response to varying bath conditions. **e** pH-dependent activation of $I_{TRPM7}$ in KO BMDMs using perforated-patch electrophysiology. Left and Right panels are as described in panel **d**. Source data are provided as a Source Data file, and statistical testing is described in "Statistics and Reproducibility".

any increase in current at pH 5.5 (Fig. 7e). These recordings indicate TRPM7 channel is activated by acidic luminal pH, which robustly elicits a monovalent current and $Ca^{2+}$-influx, most likely associated with phagosome acidification. The mechanistic implications are addressed in the Discussion section.

**TRPM7 mediates periphagosomal elevations in cytosolic $Ca^{2+}$ during efferocytosis.** Given that TRPM7 emerged as a candidate to regulate efferocytosis through a screen for $Ca^{2+}$ channels, genetically guided evidence herein that TRPM7 regulates phagosome maturation, and the known role of TRPM7 to mediate macrophage $Ca^{2+}$ signaling in response to immunomodulatory cues[20], we directly

tested the hypothesis that TRPM7 mediates $Ca^{2+}$ elevations during macrophage phagocytosis using mice expressing genetically encoded $Ca^{2+}$ indicators. We generated GCaMP6s-expressing mouse strains with ($Trpm7^{fl/fl}GCaMP6s^{CX3CR1\ Cre}$; herein, "KO GCaMP6s") and without deletion of $Trpm7$ (WT GCaMP6s) (Fig. 8a). To validate GCaMP6s function, we tested $Ca^{2+}$ responses in WT and KO GCaMP6s BMDMs to ATP. ATP is a crucial 'find me' signal secreted by apoptotic cells[24] and tunes phagocytes for efficient phagocytosis[47]. At low concentrations used in this experiment, ATP signals through Gq-coupled P2Y receptors and mediates SOCE, without activating P2X7 ion channels present on macrophages—these require upwards of 0.5 mM ATP for activation. With 2 mM $Ca^{2+}$ outside

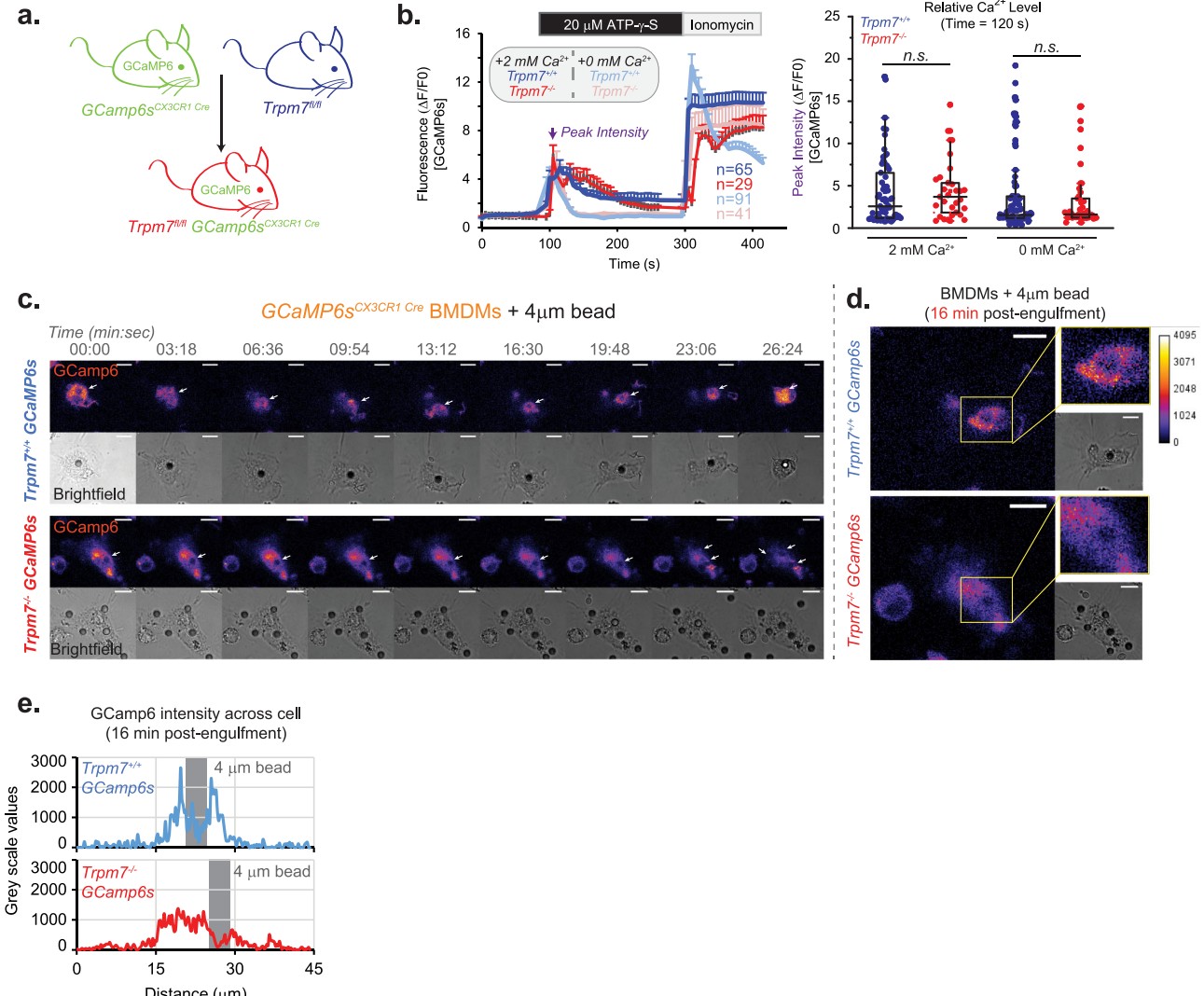

**Fig. 8 Phagocytosis of apoptotic cells triggers phagosome-proximal elevations in Ca²⁺. a** Generation of TRPM7 WT and KO GCaMP6s-expressing mouse strains is schematized. **b** Relative changes in [Ca²⁺]i, as depicted by GCaMP6s-fluorescence, in response to ATP over time in GCaMP6s-expressing BMDMs. Cells were treated with 20 μM ATP-γ-S for 3 min in bath solution containing either 0 or 2 mM Ca²⁺, as indicated. Ionomycin (2 μM) was added as a positive control. Left: Mean GCaMP6s intensity over time (n values shown in the figure); error bars represent SEM. Right: Quantification of peak GCaMP6s-fluorescence intensities after ATP-γ-S addition, at the timepoint indicated by an arrow in the left panel; Each data point in the box chart reflects a cell. The mean value is depicted by a solid horizontal line across the box. **c** Changes in [Ca²⁺]i during phagocytosis in GCaMP6s-expressing WT (top) and KO BMDMs (bottom) during phagocytosis of 4 μm carboxylated beads. Arrow indicates bead-containing phagosome. GCaMP6s-fluorescence (top image; Fire LUT) and brightfield (bottom image; gray) were acquired via confocal microscopy and single optical section (0.5 μm) is shown. Scale bar is 10 μm. **d** Periphagosomal [Ca²⁺]i in WT and KO BMDMs is shown as a magnified image from panel **a**, 16 min post-engulfment. GCaMP6s-fluorescence (fire) and brightfield (gray) are shown with ROI highlighted (yellow box). Relative scale of GCaMP6s intensity is show on the right. Scale bar is 10 μm. **e** Measurement of [Ca²⁺]i derived from a line scan analysis from cells depicted in panel **b**. Fluorescent intensity of GCaMP6s was measured across the length of the cell through the indicated phagosome in a single x–y plane line trace. Bead location indicated by gray column. Source data are provided as a Source Data file, and statistical testing is described in "Statistics and Reproducibility".

(extracellular), both WT and KO GCaMP6s BMDMs responded rapidly to 20 μM ATP-γ-S, a non-hydrolyzable analog of ATP, with a five-fold increase in mean GCaMP6s-fluorescence relative to baseline Ca²⁺ levels (Fig. 8b). ATP-triggered Ca²⁺ elevations decayed slowly, likely due to SOCE, and cytosolic Ca²⁺ concentration [Ca²⁺]i decayed toward resting levels (baseline) after nearly 220 s. Finally, the addition of ionomycin elicited the maximal Ca²⁺ response. In 0 mM extracellular Ca²⁺, the release of intracellular stored Ca²⁺ from the ER was comparable in amplitude to the SOCE seen in 2 mM Ca²⁺, but unsurprisingly, [Ca²⁺]i decayed to baseline in only 45 seconds. The addition of 2 mM extracellular Ca²⁺ and ionomycin revealed maximal Ca²⁺ elevations that were comparable to those in 2 mM

extracellular Ca²⁺. Importantly, ATP-induced Ca²⁺ elevations in both WT and KO GCaMP6s BMDMs were nearly identical. These results demonstrate the Ca²⁺-signaling in response to ATP, an important "find me" signal during efferocytosis, is completely normal in *Trpm7*-deficient macrophages.

Using live-cell confocal microscopy, we measured [Ca²⁺]i during phagocytosis of 4 μm carboxylated beads in WT and KO GCaMP6s BMDMs. Both WT and KO GCaMP6s BMDMs readily associated with the beads, and by 10 min, had started to form nascent phagosomes (Fig. 8c). In both WT and KO phagocytes, we observed modest elevations in [Ca²⁺]i levels during phagocytosis. However, WT BMDMs displayed striking

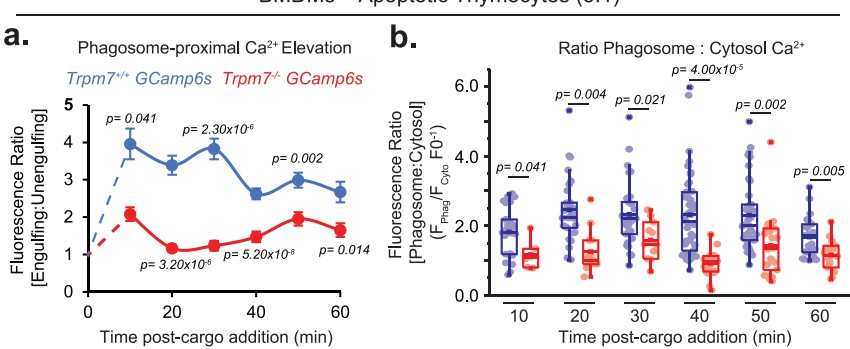

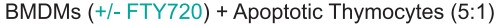

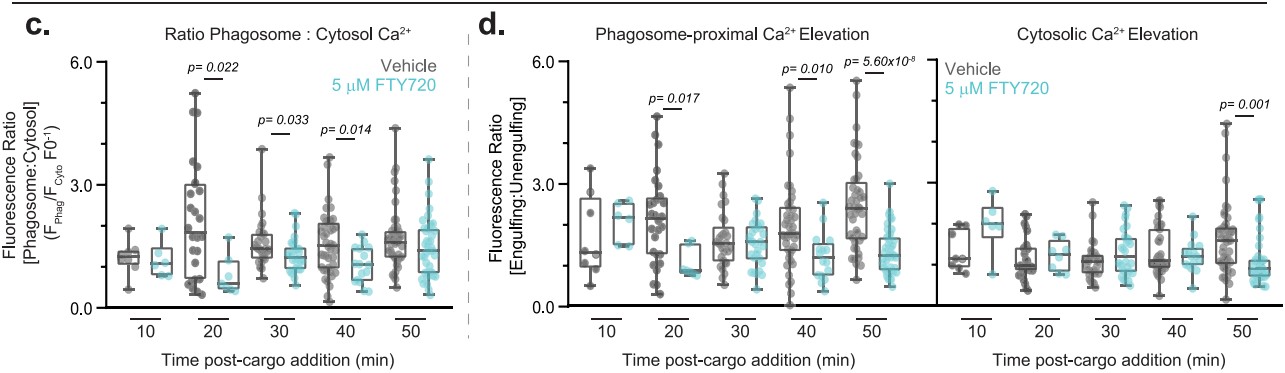

**Fig. 9 Phagosome-proximal elevations in Ca²⁺ are dependent on TRPM7 during efferocytosis. a** Measurement of phagosome-proximal [Ca²⁺] in WT and KO BMDMs during phagocytosis of apoptotic cells. Changes in fluorescence of GCaMP6s represent changes in phagosome-proximal Ca²⁺. Data points were collected at indicated time points and measure the MFI in cargo-containing BMDMs relative to the fluorescence of non-engulfing BMDMs in the same field of view. Data points depict mean values; error bars reflect SEM ($n > 20$ cells at each timepoint). Individual data points are plotted in Supplementary Fig. 6e (left), and statistics were tested at each measured timepoint between genotypes. **b** Relative changes in phagosome-proximal [Ca²⁺] during phagocytosis in WT and KO BMDMs. Phagosome-proximal values are relative to the cytoplasmic [Ca²⁺] within the same cell. Data were measured from the same experiments shown in panel **a**. Fluorescence ratio reflects fluorescence intensity in an ROI proximal to phagosome relative to the cytosol ROI. Analysis is described in Supplementary Fig. 6a-c and Methods. Data points in the box chart represent individual phagosomes and the mean value is depicted as a horizontal line across the box. See also Supplementary Fig. 6d, e. **c** Relative changes in phagosome-proximal cytosolic [Ca²⁺] during phagocytosis with and without treatment with FTY720. Data are as displayed in Fig. 9b. **d** Relative changes in phagosome-proximal and cytosolic [Ca²⁺] during phagocytosis with and without treatment with FTY720. Data are displayed as in Fig. 6f, data points in the box chart represent individual phagosomes and the mean value is depicted as a horizontal line across the box. See also Supplementary Fig. 9c. Source data are provided as a Source Data file, and statistical testing is described in "Statistics and Reproducibility".

elevations in Ca²⁺, proximal to the phagosome (Fig. 8d, e). In contrast, KO BMDMs displayed spatially abnormal elevations in cytosolic Ca²⁺ which were not concentrated around the phagosome.

We then measured [Ca²⁺]ᵢ during efferocytosis of fluorescently labeled apoptotic cells. To evaluate subcellular changes in free cytosolic Ca²⁺, GCaMP6s fluorescent intensity was measured in specific regions of interest (ROI) located in the cytoplasm and proximal to the phagocytic cargo (within 100 nm); baseline Ca²⁺ levels were measured in BMDMs that were not associated with cargo (Supplementary Fig. 6a-c). Efferocytosis initiated generalized [Ca²⁺]ᵢ elevations independently of TRPM7 (Fig. 9a, b), but in WT GCaMP6s BMDMs, phagosome-proximal Ca²⁺ was significantly upregulated during phagocytosis, peaking at 2.6-fold above cytosolic Ca²⁺ levels between 30 and 40 min after the initiation of phagocytosis. These periphagosomal Ca²⁺ elevations were maintained for several minutes. In contrast, in KO GCaMP6s macrophages, the phagosome-proximal Ca²⁺ showed relatively minor elevations above cytosolic Ca²⁺ levels (average elevation of 1.1-fold increase relative to cytosolic Ca²⁺) (Supplementary Fig. 6d, e). Similarly, in FTY720-treated BMDMs, phagosome-proximal Ca²⁺ elevations were significantly

decreased throughout phagosome maturation (Fig. 9c, d and Supplementary Fig. 6f), consistent with the deficits in phagosome acidification (Fig. 7a). Based on this evidence, we propose a model wherein TRPM7 mediates phagosome-proximal Ca²⁺ elevations during phagocytosis.

## Discussion

This study advances our mechanistic understanding of efferocytosis by identifying TRPM7 as a crucial ion channel controlling phagosome maturation through a combination of Ca²⁺-signals and pH-activated cationic flux. We show that spatiotemporal regulation of Ca²⁺-signaling is essential for subsequent phagosome maturation and efficient acidification of the engulfed cargo in the phagolysosomes. In this context, we identified a TRPM7-mediated Ca²⁺-signaling pathway that controls phagosome maturation during efferocytosis. The absence of TRPM7 compromises phagosome maturation and $Trpm7^{-/-}$ macrophages fail to digest engulfed apoptotic cells normally; these observations are corroborated by complementary approaches using siRNA and pharmacology to acutely target TRPM7. Historically, the study of intracellular Ca²⁺-dynamics in phagocytosing macrophages has been challenging because myeloid cells

are particularly efficient at pumping out small molecule $Ca^{2+}$-indicators and the use of broad-spectrum pump blockers such as Probenecid interferes with a variety of ion channels and transporters[27]. We circumvented that technical hurdle by generating mouse lines that express GCaMP6s in myeloid cells with a conditional deletion of *Trpm7*. We show that the TRPM7 channel mediates $Ca^{2+}$ elevations proximal to the phagosome during phagocytosis, and we propose that these elevations control $Ca^{2+}$-dependent membrane fusion events, such as recruitment of vesicles and phagosome-lysosome fusion, to facilitate phagosome maturation.

TRPM7 resides in macrophage cell membranes where it elicits large outwardly rectifying currents by conducting monovalent and divalent cations including $Ca^{2+}$ and $Zn^{2+}$ [38]. The counter-ion conductance of cations is necessary to sustain the $H^+$ pumping activity of the V-ATPase complex[48]. Similar mechanisms are necessary for myeloid cells to maintain endocytic integrity during macropinocytosis[49], which are regulated by two-pore channels that mediate $Na^+$ efflux necessary for endomembrane homeostasis, but the unavailability of highly selective $Na^+$ dyes retained by macrophages prevents direct subcellular visualization of this cationic countercurrent during phagosome maturation. Assembly of the mature phagosomal complex requires spatiotemporal coordination of seemingly all cellular organelles, which may require proximal localization (e.g., lysosomal fusion) or distal activity (e.g., transcriptional adaption) to support phagosome maturation. During efferocytosis, we detect TRPM7 proximal to the phagosomal membrane (Fig. 6c), but it is not yet clear whether TRPM7 is incorporated from the plasma membrane or whether vesicles contain TRPM7 fuse to the nascent phagosome. However, we do not observe baseline defects in lysosomal pH in *Trpm7*-deficient cells, and TRPM7 vesicles are not reported to contain $Ca^{2+}$ [38]. Identifying the cell biological role of vesicular TRPM7 and understanding the heterogeneity of cellular endosomes remains an objective of future studies.

Using perforated-patch electrophysiology configuration, which retains the structural integrity of the intracellular protein complexes, we show that TRPM7 channel activity in primary macrophages is highly sensitive to low pH (Fig. 7b-e). Although the low pH-evoked current was inhibited by $Mg^{2+}$, FTY720, and genetic deletion of *Trpm7*, other ion channels regulated by pH, such as PAC[50], likely play important roles in pH-sensing by macrophages. Initial acidification of the phagosome to pH ~5.5, which can occur minutes after engulfment[51], is sufficient to maximally activate TRPM7 channel, and would sustain TRPM7 activity, resulting in a large cationic discharge from the lumen to counter the V-ATPase-mediated injection of $H^+$ into the endo-phagosome. Thus, it is reasonable that pH-dependent activation of TRPM7 mediates a cationic countercurrent necessary for maintaining the proton-pumping activity of the V-ATPase in the maturing phagosome. Parsimoniously, TRPM7 may provide regulatory feedback for the maintenance of low phagosomal pH by mediating cationic flux (including $Ca^{2+}$) from the phagosome into the cytosol; implications of the $Ca^{2+}$ source are discussed later. This model is supported by the observation that *Trpm7*-deficient macrophages fail to elevate phagosome-proximal $Ca^{2+}$ levels during phagocytosis.

Alternatively, before the phagosome is sufficiently acidified, rapid changes in the phospholipid content of the phagosomal membrane could activate TRPM7. Elegant studies of FcR-mediated apoptosis have revealed that the activation of PLCγ at the phagosome rapidly depletes 4,5-$PIP_2$ in the phagosomal membrane[52–55]. The hydrolysis of $PIP_2$ activates TRPM7[41], and in our model, the resulting $Ca^{2+}$-flux from the phagosome to the cytosol promotes maturation. It is important to note that changes in phospholipid content of the phagosomal membrane have not

been studied in the case of efferocytosis, and the idea that *effer-ophagosome*-resident TRPM7 is activated by such mechanisms is currently a matter of conjecture.

Phagosomal ionic currents presume the adequate availability of free ions in the phagosomal lumen, and phagosomal $Ca^{2+}$ has been implicated as a key $Ca^{2+}$ source during macrophage phagocytosis, promoting remodeling of the phagosome and phagolysosome integration[11,14,56]. However, the source of $Ca^{2+}$ can be a point of contention because at least in the case of Fc-receptor mediated phagocytosis, the phagosomal membrane is seen tightly enveloping the opsonized cargo, leaving little fluidic volume in the phagosome to support ionic currents through the channels in the phagosomal membrane. In other studies[57], the phagosome is clearly seen to contain "cargo-free" volume that may provide an important source of $Ca^{2+}$ during phagosome maturation—this model has also been suggested by others[11,56,58]. The specific nature of the cargo and phagosomal contents likely plays a major role in determining the precise ultrastructural nature of the phagosome[59]. Although we did not measure the phagosome contents directly, these studies strongly suggest that the phagosome contains a modest residual volume in excess of the cargo, and this fluidic cargo may contribute $Ca^{2+}$ currents from the phagosome into the cytosol. The precise measurements of nascent phagosomal $Ca^{2+}$ concentrations face major technical hurdles because of the relative sensitivity of $Ca^{2+}$-indicators to pH. In the later stages of phagosome maturation, the presence of certain lysosomal channels, such as TRPML1 in the lysosomes[60], may also promote the $Ca^{2+}$ signaling necessary to sustain the physiology and specialization of the phagolysosome.

Our study has focused on the role of TRPM7 during efferocytosis, and future studies will investigate whether TRPM7 is also crucial in other forms of phagocytosis. In particular, defining the role of TRPM7 in FcR-mediated phagocytosis of antibody-opsonized cells (e.g., virally infected cells that display viral coat proteins) and in the phagocytosis of pathogens capable of activating pattern-recognition receptors (e.g., Toll-like Receptors (TLRs), Mannose receptor, Dectin-1 etc.). Likewise, the siRNA screen also revealed additional ion channels that regulate phagosome maturation during efferocytosis, and independent studies are ongoing to test their role in phagosome maturation—these channels likely play exciting roles in further coordinating the spatiotemporal concert of organelle and $Ca^{2+}$ signaling during phagocytosis. Much of the previous work on phagocytosis-associated $Ca^{2+}$-signaling has been carried out in the context of FcR-mediated phagocytosis, which triggers a robust SOCE response. Similarly, ATP-mediated activation of P2Y receptors is a potent "find me" signal for apoptotic cells[24,25]. Our expectation was that SOCE, likely initiated by Gq-coupled P2Y receptors in response to purinergic signals, would play a major role in engulfment or phagosome maturation during efferocytosis. Surprisingly, the disruption of SOCE apparatus through the simultaneous depletion of *Stim1* and *Stim2* fails to have a significant negative impact on phagosome acidification, and efferocytosis appears to proceed normally. It should be noted that the deletion of *Stim1* and *Stim2* does not prevent the release of ER-stored $Ca^{2+}$, so our observations do not challenge the importance of $Ca^{2+}$-release from the ER during phagocytosis[61,62]. Indeed, even in *Trpm7*$^{-/-}$ macrophages, we see substantial $Ca^{2+}$ elevations in the cytosol during efferocytosis, and we speculate that these are largely mediated by the release of stored $Ca^{2+}$. We did not observe any effect of deletion of *Trpm7* on ATP-triggered release of ER-stored $Ca^{2+}$ or the subsequent SOCE. Therefore, it is possible that TRPM7-mediated periphagosomal $Ca^{2+}$-signaling and ER-mediated global $Ca^{2+}$signaling cooperate to orchestrate the $Ca^{2+}$-signaling processes during phagocytosis. An interesting implication is that different kinds of phagocytosis may show

varying reliance on different modes of $Ca^{2+}$-signals[63], with efferocytosis more reliant on TRPM7 and FcR-mediated phagocytosis of opsonized targets more reliant on SOCE; these independent studies are ongoing.

This study focused on the role of TRPM7 channel, but whether TRPM7 kinase contributes meaningfully to phagosome maturation remains unknown. TRPM7 kinase may regulate actomyosin by phosphorylating the Myosin II Heavy Chain and it can also phosphorylate Annexin A1, a modulator of membrane curvature[64,65]. The TRPM7 kinase may also regulate the recruitment of other proteins associated with the degradation of apoptotic cell cargo[66]. The proteolytic cleavage of TRPM7 is known to activate the channel and liberate a fully functional kinase domain that is untethered to membrane-resident channel[39,67]. It is not known whether such cleavage occurs during efferocytosis.

Ultimately, a fusion of the phagosome with lysosomes forms an acidic organelle called phagolysosome, wherein, the acid-optimized lysosomal degradative enzymes complete the task of fully processing the engulfed cargo in a timely manner[68]. Interestingly, the uptake of non-opsonized apoptotic cells results in a significantly faster rate of phagosome maturation, when compared to Fc-receptor mediated engulfment of opsonized dead cells[69]. This observation supports the notion that mechanisms and kinetics of phagosome maturation may vary significantly between different types of phagocytosis[70], and these differences may have major implications on what constitutes non-inflammatory clearance of dead cells, versus a mechanism that activates an immune response to a pathogenic or an antibody-labeled immunogenic cargo. Thus, the control and timing of phagosome maturation represent a crucial checkpoint at the crossroads of inflammation and tissue homeostasis.

In addition to the cell biological questions outlined above, an important goal of our future studies is to define the implications of abnormal phagosome maturation in the context of innate immunity, tissue homeostasis, and wound repair. A hallmark of efferocytosis is that the clearance of apoptotic cells and debris occurs in a non-inflammatory manner. However, in the absence of TRPM7, the cell-autonomous response to the abnormal digestion of the apoptotic cell cargo could be very different—these studies are likely to reveal profound insights that are of significance to many age-related diseases where the control of inflammation and tissue regeneration is of salience.

## Methods

**Mouse strains**. Mice used followed procedures approved by the Animal Care and Use Committee of the University of Virginia and adhered to National Institutes of Health Animal Care and Use Guidelines and. Mice were housed in 12 h light-dark cycle at room temperature (20–23 °C) with 40–70% humidity with ad libitum access to food and water. Male and female mice aged 7–14 weeks were used for all experiments. *Trpm7fl/fl* and *Trpm7fl/fl (LysM cre)* mice were generated as described previously. *Trpm7fl/fl (CX3CR1 Cre)* mice were generated through crossing *Trpm7fl/fl* mice to B6J.B6N(Cg)-Cx3cr1tm1.1(cre)Jung/J (Jackson Laboratories; 025524) to subsequently generate *Trpm7fl/fl* and *Trpm7fl/fl (CX3CR1 Cre)* mice on a mixed background. GCaMP6s-expressing strains were generated crossing established mouse strains with the B6;129S-Gt(ROSA)26Sortm96.1(CAG-GCaMP6s)Hze/J mice, which contain a LoxP-flanked STOP codon. Deletion of *Trpm7* was confirmed via quantitative real-time PCR analysis and patch-clamp electrophysiology.

**Cell culture**. Bone-marrow-derived macrophages were isolated and cultured as previously described[71]. In brief, bone marrow was extracted from the murine femur and tibia via centrifugation. Bone marrow was then lysed in ACK Lysis buffer and plated on petri dishes in BMDM Media (RPMI 1640 + 10% FBS + 20% L929-conditioned media). Cells were differentiated for 7 days and the media was refreshed afterwards every 3 days. For experiments, BMDMs were used between day 7 to 14 post-differentiation. For cultured cell lines, Jurkat (Clone E6-1 (ATCC® TIB-152™)) and RAW 264.7 (ATCC® TIB-71™) cells were cultured and maintained according to the vendor's instructions. Jurkat-GFP cells were a gift from Dr. Kodi Ravichandran (University of Virginia, presently WashU in St. Louis) and maintained in T75 culture flasks (RPMI + 10% FBS) at 0.1–2.0 × 10⁶ cells/ml. For

experiments using BAPTA-AM or EGTA-AM, cells were loaded for 30 min at RT in serum-free media to minimize accumulation in vesicles and organelles; media was replaced with warm culture media prior to experimentation. Cytochalasin D was added in culture media at 37 °C for 15 min prior to experimentation. Bafilomycin A1 was added simultaneously with cargo.

**Preparation of apoptotic cells**. Thymocytes were freshly isolated prior to experimentation. Briefly, mice were euthanized, the thymus was removed, and cells were dissociated through a 40 μm cell strainer in PBS. Thymocytes were collected via centrifugation ($300 \times g$ for 5 min), resuspended in ACK lysis buffer for 5 min at RT, pelleted via centrifugation, and resuspended in HBSS. In experiments using Jurkat cells, cultured Jurkat cells were collected via centrifugation and resuspended in HBSS. Cell suspensions were added to petri dishes, which were irradiated with 150,000 μJ of UV light (UV Stratalinker 1800). Cells incubated for 3 h at 37 °C and 5% $CO_2$. After 3 h, apoptotic cells were pelleted by centrifugation ($350 \times g$ for 5 min) and resuspended in HBSS. Cells were stained with indicated fluorescent dyes for 30 min at 37 °C. For flow cytometry, where indicated, cells were stained with CypHer5E NHS Ester (GE Healthcare; PA15405) and CellTrace Violet (Thermo Fisher; C34557), each at 1 μM. Apoptotic cells were washed 3× in culture media (containing 10% FBS) and destained by incubating in media for 30 min at 37 °C. Apoptosis was confirmed by Annexin V-7AAD viability staining (BD Biosciences; Cat# 559763) and measured via flow cytometry. Cells were pelleted and resuspended in culture media for counting "total" numbers of cells via trypan blue exclusion. Apoptotic cells were added to macrophages at the ratio indicated in the figures and legends. Reagents are also listed in Supplementary Table 1.

**Measurement of phagocytosis by flow cytometry**. Macrophages were plated on 24-well non-coated tissue culture plates at a density of $0.1 \times 10^6$ cells/well overnight prior to experimentation. Apoptotic cells were prepared as described above; beads were washed 3× in culture media prior to experimentation. For pharmacological treatments, macrophages were incubated with the drug at the indicated concentration for 30 min prior to the addition of cargo in serum-free media and added with cargo in culture media. After the addition of cargo, plates were then centrifuged at $100 \times g$ for 1 min, rotated, and centrifuged for an additional 1 min to synchronize phagocyte-cargo contact. Plates were incubated at 37 °C and 5% $CO_2$ for indicated time points. At the end of the experiment, cells were washed 3× with cold PBS and incubated with 0.05% trypsin-EDTA for 5 min to remove unengulfed cargo and detach macrophages. Cold culture media was added to inhibit trypsin activity and cell suspensions were then collected ($300 \times g$ for 5 min), washed 1× with PBS, and resuspended in FACS Buffer. With Jurkat cargo, mCD45-FITC (Clone: 30-F11; 0.5 μg/ml) was used to further discriminate macrophages from cargo; with thymocytes, CD11b-Alexa fluor 488 (Thermo Fisher; Cat# 53-0112-80; 0.5 μg/ml). Instrument voltages and gating were established based on unstained macrophages, unstained cargo, and single-stained fluorescent controls (with 0.01% HCl added for CypHer5E positive control). The gating strategy is described in Supplementary Figures. Measurements were acquired using the Attune NxT (Thermo Fisher) and analyzed using FlowJo (BD Biosciences).

**$Ca^{2+}$ channel screen using siRNA**. On day 7 post-isolation, BMDMs were collected and aliquoted into individual cell suspensions for each siRNA tested. BMDMs were resuspended in OPTIMEM with appropriate SMARTpool siRNA (Dharmacon) and Lipofectamine 3000 (ThermoFisher; L3000015), according to the manufacturer's instructions. Cells were plated at $0.5 \times 10^6$ cells/mL in six-well non-treated culture plates for 48 h. After 48 h, cells were washed 3× with HBSS and detached using 0.05% trypsin. Cells were then counted via trypan blue exclusion and normalized to $0.3 \times 10^6$ cells/tube/siRNA. Cells were resuspended in siRNA-containing transfection media, as described above, and plated at $0.1 \times 10^6$ cells/well on 24-well non-treated culture plates in culture media for 48 h. On the day of the experiment, cells were washed 1× with PBS and resuspended in culture media. Gene knockdown was confirmed via quantitative real-time PCR[20]. Z-scores were calculated as the difference between CypHer MFI of *siScramble*-treated cells from the target gene divided by the standard deviation of the data set; z-scores across the experiment were then averaged and plotted as shown. All sequences of qPCR primers (Supplementary Table 2) and siRNA (Supplementary Table 3) are provided in the Supplementary Materials.

**Perforated-patch-clamp electrophysiology**. TRPM7 currents ($I_{TRPM7}$) were measured using perforated-patch configuration using similar methods described previously[41] and depicted in Fig. 5e The extracellular bath solution contained (in mM): 140 NaCl, 5 KCl, 1 MgCl₂, 2 CaCl₂, 10 HEPES, 10 Glucose (adjusted to pH 7.3 with osmolality 280 mOsm/kg). The pipette solution contained (in mM): 120 K-methanesulfonate, 30 KCl, 1 MgCl₂, 0.2 CaCl₂, 1 EGTA, 10 HEPES (adjusted to pH 7.3 with osmolality 273 mOsm/kg). A final concentration of 400 μg/ml Nystatin was used to establish membrane perforation and prepared by adding 16 μl of Nystatin dissolved in DMSO to 1 ml of pipette solution; this was prepared fresh for each experiment. BMDMs were freshly plated prior to analysis. MgCl₂ (10 mM) or FTY720 (5 μM) was added to the external solution to inhibit $I_{TRPM7}$. For recordings, a ramp from −100 to +100 mV over 400 ms was used, with a holding potential of 0 mV; signals were low-pass filtered at 5 kHz and sampled at 10 kHz.

All electrophysiology recordings were conducted at RT with an Axopatch 200B amplifier (Molecular Devices, Sunnyvale, CA) using pClamp (v9) software.

**Immunocytochemistry.** Cells were plated overnight on coverslips prior to experiments. Following experimentation, coverslips were washed 3× in PBS to remove media and unbound apoptotic cells. For LysoTracker Red staining, cells were stained in PBS with LysoTracker Red (250 nM) at RT for 15 min. Coverslips were fixed in 4% PFA (in PBS) for 30 min at room temperature (RT). In certain experiments (Fig. 6c), coverslips were pretreated with Na-borohydride (0.1% in PBS) or TrueBlack (Biotium; Cat. #23007; 1× according to the manufacturer's instructions) to remove background autofluorescence; both pretreatment regimens were comparable in their ability to reduce autofluorescence. Coverslips were washed 3× in wash buffer (PBS with 0.05% Tween-20), blocked at RT for 1 h in blocking buffer (5% donkey serum, 1% BSA, 0.1% fish gelatin, 0.1% Triton X-100, and 0.05% Tween-20 in PBS), and incubated with primary antibody diluted in blocking buffer overnight at 4 °C. Coverslips were washed 3× in wash buffer and incubated at RT with the appropriate secondary antibody in blocking buffer for 90 min, followed by 3× wash in wash buffer. If stained for F-actin, coverslips were stained with fluorophore-conjugated phalloidin in wash buffer for 15 min at RT and washed 1× with wash buffer. Coverslips were then mounted on glass slides (ProLong Gold Antifade; ThermoFisher #P36930), allowed to cure overnight, and imaged within 48 h. Confocal microscopy was performed on an Olympus Fluoview FV1000 and Zeiss LSM880. Data were acquired with Olympus Fluoview (Ver 4.1a) or Zen (Zeiss) software and analyzed using ImageJ. All images within an experiment were acquired under identical conditions; each image was processed identically across the entire image with minimal adjustments to brightness and contrast.

**Ca²⁺ imaging during phagocytosis.** GCaMP6s-expressing BMDMs were plated on coverslips overnight prior to imaging. Coverslips were washed 1× in culture media. For apoptotic cell cargo, cells were stained with 2.5 μM TAMRA-SE or 1 μM CypHer5E (in RPMI 1640) for 30 min at RT followed by 3× wash in RPMI 1640 + 10% FBS to remove the unbound dye and placed in imaging media (RPMI 1640 without phenol red + 20 mM HEPES). For phagocytosis of polystyrene beads, 4 μm carboxylated beads were washed in FBS followed by PBS prior to imaging. The phagocytic cargo was added at a ratio specified in figures and legends for 10 min at RT to promote association with BMDMs, but minimize engulfment. Coverslips were washed 1× in imaging media and placed in the imaging chamber. Coverslips were imaged at 37 ± 1 °C. Wide-field microscopy was performed Zeiss Axio Observer microscope using a Pinkle filter set (Semrock; GFP/dsRed 2X-A-000) and Lambda DG4 Illuminator. Fluorescence was excited using a DG4 Illuminator (Sutter Instruments, Canada) and detected using an ORCA-Flash 4.0 V2 CMOS camera (Hammamatsu). Data were acquired using SlideBook software (3i) and analyzed using Origin Pro and Microsoft Excel. Microscopy was performed on the Leica SP5 confocal microscope with excitation from 'White light' and 488 nm argon lasers using Leica Applicate Suite Software (Leica) and analyzed using ImageJ. For measurements in Fig. 6, data analysis was performed as described in Supplementary Fig. 6a-c. In brief for each image, ten ROI from at least three non-engulfing cells were used to establish baseline, "non-engulfing" fluorescence at each timepoint—this internal control minimized coverslip-to-coverslip and timepoint variability which can be introduced by comparing fluorescence across samples with non-ratiometric Ca²⁺ indicators, including GCaMP6. In engulfing cells, fluorescence was measured from ROIs proximal to the phagosome and distally in the cytosol to compare "peri-phagosomal" to "cytoplasmic" Ca²⁺ changes. These measurements allowed the calculation of relative cytosolic, phagosome-proximal, and phagosome:cytosol Ca²⁺ levels shown in Fig. 9a-d and Supplementary Fig. 6d-f.

**Measurement of phagocytosis with an expression of TRPM7 Constructs.** FLAG-TRPM7 construct was developed as previously described[39]. For Image-Stream flow cytometry, prior to transfection, RAW 264.7 cells were plated at 0.2 × 10⁶ cells/well in 12-well plate. FLAG-TRPM7 or GFP was transfected into RAW 264.7 cells using TransIT-X2 transfection reagent (Mirus) according to the manufacturer's protocol at 1 μg of plasmid/well (0.1 μg plasmid/well for GFP). Transfected RAW 264.7 cells were cultured for 24 h prior to experimentation. Apoptotic cell cargo was added as described for the assay in figures and legends, and cells were collected as described above. Phagocyte cell suspensions were then fixed (2% PFA for 60 min), permeabilized (permeabilization buffer; FACS Buffer + 0.05% Triton X-100), and permeabilization buffer with 5% FBS + FcBlock antibody (1:100) for 1 h at RT with agitation. Cells were then incubated with an anti-M2 Flag antibody (Thermo Fisher MA1-91878 or Sigma #F1804; 1:1000) for 2 h at RT, washed 3× in permeabilization buffer prior to incubation with a secondary antibody (donkey anti-mouse IgG-Alexa-fluor 488 (Thermo Fisher, #A21202)) for 2 h at RT. Following 3× washes in permeabilization buffer, cells were resuspended in FACS buffer prior to analysis via ImageStream flow cytometry. GFP-expressing cells were used as a negative control for colocalization, representing non-specific, cytoplasmic fluorescence proximal to the cargo. ImageStream data were acquired using the Amnis ImageStream (EMD Millipore) flow cytometer and analyzed using INSPIRE analysis software according to the manufacturer's

guidelines (Amnis-EMD Millipore); similarity scores were calculated by the software.

For experiments using LR73 phagocytes, cells were transfected with 1–10 μg of pcDNA6-Flag-TRPM7 using TransIT-X2 transfection reagent according to the manufacturer's protocol. After 48 h, cells were reseeded onto glass coverslips in a 24-well plate overnight. 4.0 μm FluoSpheres (Thermo Fisher, #F8858) were fed to LR73 cells at a 10:1 ratio (beads:cell) for 2 h. Cells were then fixed in 4% PFA in PBS for 15 min, and prepared for immunocytochemistry (as described in "Methods"). Primary antibody (anti-M2 Flag; Sigma #F1804 or Thermo Fisher MA1-91878; 1:1000) was added to the blocking buffer overnight at 4 °C, followed by staining secondary antibody (donkey anti-mouse IgG-Alexa-fluor 488). Coverslips were sealed with clear nail polish and imaged on Zeiss LSM880 confocal microscope.

**In vivo measurement of phagocytosis.** Male and female mice were injected intraperitoneally with 200 μl HBSS or 4 × 10⁶ apoptotic cells resuspended in HBSS. Apoptotic Jurkat-GFP cells were prepared as described herein for Jurkat cells. After 90 min, mice were euthanized and peritoneal lavage was performed as described previously. Recovered cells were pelleted, resuspended in ACK lysis buffer for 5 min (red blood cell lysis), washed 1× in cold PBS, and stained for flow cytometry. For peritoneal phenotyping, cells were stained with TruStain FcX anti-mouse CD16/32 (5 μg/ml; Biolegend; #101320) for 10 min at 4 °C prior to the addition of the fluorophore-conjugated antibody cocktail for 30 min. As shown, antibodies used for staining were anti-mouse CD11b FITC (Clone: M1/70), F4/80 PE-Cy7 (Clone: BM8), and CD11c APC (Clone: N418) at 0.5 μg/ml. Cells were then washed 2× in FACS Buffer (0.5% BSA, 2 mM EGTA) prior to analysis. All experiments included single stain controls using OneComp eBeads (Thermo Fisher; #01-1111-42) to determine appropriate compensation values and fluorescence minus one control with cells for gating. Flow cytometry measurements were performed on the Attune NxT Flow Cytometer (Thermo Fisher) and analysis performed in FlowJo Software (v10; FlowJo, LLC BD). Gating strategies are shown in Supplementary Figs., and additional reagent information is in Supplementary Table 1.

**Generation of DQ green BSA reporter beads.** DQ Green BSA reporter beads were prepared as described previously, with minor modifications[36]. In brief, 5 × 10⁷ carboxylate-modified 4 μm beads were washed 3 times in 1 mL of PBS (2000 × g for 2 min) and vortexed to disperse into 'single-bead' suspension in PBS (pH 7.2). Freshly-prepared solution of cyanamide (crosslinker) was added to a final concentration of 25 mg/ml and incubated for 15 min with agitation. Beads were then washed twice in coupling buffer (0.1 M Na-borate in sterile H₂O; pH 8.0 with NaOH) and resuspended in 1 mg/ml DQ Green BSA in coupling buffer for 12 h at 4 °C. Beads were then washed twice in 250 mM glycine (in PBS; pH 7.2) followed by two washes in PBS. Beads were washed in complete media prior to experimentation.

**Ca²⁺ imaging with ATP stimulation.** GCaMP6s-expressing BMDMs were plated on #1 coverslips overnight prior to imaging. Cells were imaged in Ringer Solution ([in mM] 155 NaCl, 4.5 KCl, 2 CaCl₂, 1 MgCl₂, 5 HEPES, 10 glucose, pH 7.4) or in Ca²⁺-free Ringer Solution (155 NaCl, 4.5 KCl, 5 EGTA, 1 MgCl₂, 5 HEPES, 10 glucose, pH 7.4) at RT. Measurements were performed with perfusion of bath solution with or without ATP-γ-S (20 μM) using a gravity feed system and ionomycin (2 μM). Fura-2-AM measurements were performed as described previously[20] with stimulations as indicated in the figure legend. Fluorescence was excited using a DG4 Illuminator (Sutter Instruments, Canada) and detected using an ORCA-Flash 4.0 V2 CMOS camera (Hammamatsu) using SlideBook 6 software (3i).

**Lysosomal activity assays.** For LysoSensor experiments, BMDMs were stained with LysoSensor Green DND-189 (Thermo #L7535) according to the manufacturer's protocol. Briefly, BMDMs were incubated with LysoSensor (1 μM) in culture media for 30 min at 37 °C prior to collection. Macrophages were then washed in PBS, scraped off the plates, and resuspended in FACS buffer for acquisition on Attune NxT Flow Cytometer (Thermo Fisher). For DQ Green BSA proteolysis, BMDMs were incubated with 10 ug/ml DQ Green BSA in a complete medium at 37 °C for 30 min and 60 min. DQ-BSA containing medium was then washed out with 1× PBS, cells scraped and resuspended in FACS buffer for acquisition on Attune NxT (Thermo Fisher) and analyzed using FlowJo (BD Biosciences).

**Statistics and reproducibility.** All data were analyzed using GraphPad Prism 8.0 or 9.0 (GraphPad Software), Origin Pro 9.1.0 (Origin Lab), or Excel (Microsoft) software. Data are presented as individual data points of independent samples or means with error bars as indicated in figure legends. Bar charts and line graphs were plotted in Excel; dot plots and box charts, GraphPad Prism; box charts, Origin Pro. Statistical box charts contain boxes and whisker bars, which denote the 25–75 and 1–99 percentile range, respectively, with individual data points overlaid; the median value is denoted by a horizontal line and the mean by a rectangle. Normality was tested graphically (QQ plots) or D'Agostino-Pearson normality tests. Data that passed normality and variance tests were analyzed by t-tests for 2 groups

and ANOVA for >2 groups, all using two-tailed tests; Bonferroni post-hoc correction for multiple comparisons was applied where appropriate. Mann-Whitney test was used for non-parametric data with two groups. Sample sizes are indicated in figures and legends, and $p$-values less than 0.05 were considered statistically significant (* denotes $p < 0.05$; **$p < 0.01$; ***$p < 0.001$; ****$p < 0.0001$).

**Reporting summary**. Further information on research design is available in the Nature Research Reporting Summary linked to this article.

## Data availability

Custom code was not used in this study. The IMMGEN data used for the siRNA screen are freely accessible at https://www.immgen.org/. The methods, reagents, targets, and verified hits from this study are contained in the manuscript. In addition to the source data file, data in this manuscript and part of ongoing studies are also available upon request to the corresponding author. Source data are provided with this paper.

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

## Acknowledgements

We thank members of the Desai lab for their insightful comments. We thank Dr. K. Ravichandran (UVA, currently WashU) and lab members for their assistance with reagents and helpful discussions. The following core facilities provided technical resources and support: KECK Cellular Imaging Center, UVA Flow Cytometry Core, UVA Advanced Microscopy Facility, Carter Immunology Center Flow Cytometry Core, and UVA Cardiovascular Research Center Microscopy Facility. We also thank our funding: B.N.D. (GM108989 and AI155808), M.S.S. (5T32GM007055), and T.K.D. and M.S.S. (UVA Double Hoo Award).

## Author contributions

Conceptualization: M.S.S. and B.N.D.; Methodology: M.S.S., M.E.S., T.K.D., S.K.M., and B.N.D.; Software: n/a, Validation: M.S.S., M.E.S., G.W.B., T.K.D., S.K.M., P.V.S., Z.J.F., and B.N.D.; Formal analysis: M.S.S., G.W.B., T.K.D., S.K.M., Z.J.F., M.E.S., and P.V.S.; Investigation: M.S.S., G.W.B., T.K.D., S.K.M., Z.J.F., M.E.S., P.V.S., and C.A.D.; Resources: T.K.D., E.J.S., M.E.S., and P.V.S.; Data curation: M.S.S., G.W.B., M.E.S., T.K.D., S.K.M., P.V.S., Z.J.F., and B.N.D.; Writing—original draft preparation: M.S.S. and B.N.D.; Writing—review and editing: M.S.S., M.E.S., and B.N.D.; Project Administration: B.N.D.; Funding acquisition: B.N.D.

## Competing interests

The authors declare no competing interests.
