## [Peer Review File · Nature Communications]

Reviewers' comments:

Reviewer #1 (Remarks to the Author):

Unclear are the spatiotemporal calcium responses of apoptotic engulfment by macrophages, in contrast to studies of Fc-receptor mediated phagocytosis.

Stimulated bone marrow-derived macrophages largely fail to acidify phagosomes when incubated with fast calcium chelator BAPTA. This is in contrast to slow Ca-chelator EGTA-AM and appears independent of basal modulation of lysosomal function. This is a resourceful approach to tackle the challenging question of calcium flux in macrophages and consistent with the careful conclusion with caveats that Ca dynamics proximal to the phagosome are important during efferocytic engulfment.

GCaMP6s-CX3CR1-Cre mice in support exhibit Ca²⁺ oscillations during initial stages of apoptotic cell engulfment, and TRPM7 is newly implicated after a selective screen, which was validated by percent knockdown of candidate genes. TRPM7 is required for efferocytic acidification.

It is laudable that the Investigators seek an in vivo test of the findings in culture, although the experiment relies on the infusion of apoptotic cells that were prepared in culture. Thus, is this really in vivo efferocytosis? This Reviewer also appreciates the steps to control basal effects of the Trpm7 knockout model, however the F4/80 Cd11c analysis of the peritoneal population is a relatively simplistic analysis of the heterogeneous peritoneal cell subset population. Another trivial explanation of the results in this model could potentially be ascribed to effects on expression of efferocytic receptors, which could potentially be resolved by examining levels of apoptotic to phagocyte cell binding.

Mechanisms of TRPM7 are investigated and data indicate that Trpm7 is required for phago-lysosome fusion. Interestingly, TRPM7 associates with the nascent phagosome and is activated by low pH. Efferocytosis requires TRPM7 for phagosome-proximal elevations in calcium.

Reviewer #2 (Remarks to the Author):

This study reports that the TRPM7 channel promotes the acidification and maturation of macrophage phagosomes during clearance of apoptotic cells, a process known as efferocytosis. Screening for candidate channel genes impacting phagosomal pH, the authors identify TRPM7 as a positive regulator of phagosome acidification. Bone-marrow derived macrophages (BMDM) from TRPM7-deficient mice failed to acidify and had reduced proteolytic activity. FLAG-TRPM7 expressed in a macrophage cell line was detected near phagosomes. Outwardly-rectifying currents were recorded in BMDM from WT exposed to acidic solutions that were absent in BMDM from TRPM7-deficient mice, which exhibited reduced Ca^{2+} elevations around phagosomes. The authors conclude that the TRPM7 channel sustains efferocytosis in macrophages by mediating the flux of sodium and calcium ions from the phagosome to the cytosol. They propose that the efflux of sodium ions provides a counter-current that sustains proton pumping by the V-ATPase while the efflux of calcium generates vicinal calcium elevations that promote phagosome maturation.

Comments: The study is timely, the experiments well-designed, and the data for the most part convincing. The pH recordings with CypHer5E clearly show that TRPM7 is required for phagosomal acidification. The Ca^{2+} recordings in primary macrophages from mice expressing the genetically encoded calcium indicator GCaMp6 show that TRPM7 contributes to cellular Ca^{2+} signals during phagocytosis. On the other hand the electrophysiological and morphological evidences are less solid and would require further confirmation. I also have issues with the presentation and interpretation of some data. Specifically:

1. The electrophysiological data appear preliminary. Fig. 5F shows recordings from 5 cells, while Fig. 5G and 5H show single recordings. No statistics are provided and the number of cells recorded is not mentioned. Was only one cell recorded for each genotype? These data need to be substantiated with additional recordings and statistically evaluated. The interpretation of the patch-clamp data is also questionable. The recordings show outwardly-rectifying currents reversing at 0 mV in Na^+/K^+ -based solutions, indicating that the current is carried by monovalent cations (predominantly, K^+ efflux). Yet, the authors conclude that the channel contributes to the periphagosomal Ca^{2+} elevations by conducting Ca^{2+} ions from the phagosome to the cytosol. This claim is not supported by the electrophysiological evidence, and the directionality of the flux cannot be ascertained without knowledge of the phagosomal membrane potential and of the intraphagosomal Ca^{2+} concentration.

2. The evidence that TRPM7 is present on the membrane of phagosomes is inconclusive. TRPM7 is known as a plasma membrane channel. Here the authors are proposing that TRPM7 is acting as a phagosomal calcium channel. Critical for this claim is to show that TRPM7 actually localizes to the phagosomal membrane, but the confocal images presented are unconvincing. Fig 5C and S4B show immunostaining of LR73 phagocytes expressing FLAG-tagged TRPM7, but the images do not allow a proper evaluation of the labelled structures. Higher resolution and magnification images should be

provided, after deconvolution, and the authors should make the complete stack available. Images from the stream analysis should be provided as it is impossible to judge whether that analysis is convincing without seeing the images. A control staining on untransfected cells should be provided.

Immunostaining in macrophages are notoriously difficult, as in addition to abundant Fc receptors that bind to IgG, they contain a lot of vesicles that are highly autofluorescent, especially if cells are cultured in media containing phenol red. It is unclear if some of this staining is non-specific. Pre-treating the cells with 0.1% sodium borohydride can quench autofluorescence if the authors find this is a problem. If the difference is not striking it might be good to run these untransfected but stained cells through the image stream to see how specific this colocalization signal really is. There may be an increase in vesicle clustering around phagosomes that is TRPM7-dependent, but from an action at the PM and not at the phagosome membrane. The blotchy staining in the middle of the cell could be vesicles as suggested but could also represent TRPM7 in the ER. ER cisternae at membrane contact sites with phagosomes would not be distinguishable from the phagosomal membrane at this level of resolution. Double staining with PM markers such as Cell Mask, ER marker like ER tracker, and phagosomal membrane marker such as GFP-lamp1 would document whether the TRPM7 immunoreactivity is really co-localizing with the phagosomal membrane.

3. The DQ-Green unquenching assay does not report phago-lysosome fusion, as incorrectly stated in Fig 4A and in the text on pp. 10-11. Instead, the assay reports the proteolytic capacity of phagosomes, which is greatly influenced by the ambient pH. The difference observed could thus reflect a reduced activity of acidic proteases in TRPM7 KO cells that fail to acidify. To conclude that decreased fusion of lysosomes is actually also occurring one would need to use a FRET-based approach as in Russell and colleagues (<https://doi.org/10.1002/0471142735.im1434s102>). At such, the DQ-BSA assay reflects decreased proteolytic activity, not defective fusion, and its interpretation should be revised accordingly.

4. The authors previously showed that TRPM7 mediates an influx of Ca²⁺ at the plasma membrane of macrophage that is essential for their activation by LPS and for the endocytosis of TLR4 <https://www.ncbi.nlm.nih.gov/pubmed/28130783>. Difference in the activation states of BMDM from WT and KO mice (cultured for 7 days in FBS and conditioned media) could therefore contribute to the phenotype observed. Reduced macrophage activation and endocytosis could both impact the phagosome maturation process during efferocytosis. A reduced influx of Ca²⁺ across the membrane could contribute to the reduced Ca²⁺ signals recorded in macrophages from TRPM7 cells during efferocytosis. These points should be discussed and the contribution of the Ca²⁺ entering across plasma membrane channels should be evaluated by adding channel blockers after particle engulfment.

Other points:

5. The use of quiescent cells in Figs 6 and S5 as a reference to normalize Ca²⁺ responses occurring in phagocytosing cells is questionable. First, it is not clear what this representation adds compared to the phagosome/cytosol ratio. Second, as discussed above, WT and TRPM7 null cells could have different

activation states or different Ca²⁺ influx rates, resulting in different resting Ca²⁺ levels. Finally, the degree of Ca²⁺ activity would be expected to be correlated to the phagocytic activity, yet cells with different phagocytic indexes are grouped in the “engulfing” condition. It would make more sense to compare Ca²⁺ signals occurring in cells ingesting a comparable number of phagosomes. The fluorescence images and recordings should be expressed as F/F₀.

6. Fig S1H: The apparent increase in cargo association in the BAPTA condition could reflect the pH-dependent fluorescence of the Cell Trace Violet, illustrated in Fig S1D. Please discuss.

7. The group of Reinhold Penner showed that TRPM7 regulates SOCE through its kinase domain, and that Ca²⁺ influx through TRPM7 is essential for the refilling of ER Ca²⁺ stores <https://www.ncbi.nlm.nih.gov/pubmed/28130783>. The data of Fig 6B are at odd with this finding, since they show normal long-lasting Ca²⁺ responses to an agonist in Ca²⁺ containing media. Did the authors attempt to compare the store-operated Ca²⁺ entry in WT and TRPM7 null cells? In any case, the divergence with earlier studies should be discussed.

8. Pg 9/Fig. 2C. The authors claim there are no significant differences in cargo uptake, but in the figure, there are stars above two of the three pairs of data points indicating significant differences. Is this a small but significant difference? Could the authors please clarify? What p value stars correspond to should be mentioned in the figure legend

Minor comments:

1) Pg 6 End of first paragraph – Fig. 1b should be referred to after describing the Cyt D experiment

2) Pg 8 – I was surprised to see that TRPC1, TRPC3, and TRPC6 were not included in the list of calcium permeable channels in macrophages as several studies have been published about these channels in this cell type. Please comment.

3) Pg 8/Fig1h – according to journal policy the full data set should be available - a supplementary table would be a useful way to present these data.

4) Pg 8/Supp Fig 2d/ ionomycin experiments – data should be shown, even if just in supplementary according to journal policy

5) The authors need to mention that the FTY720 compound used here to inhibit TRPM7 is also a potent agonist of S1P G-protein coupled receptors.

Reviewer #3 (Remarks to the Author):

Efferocytosis, phagocytic clearance of apoptotic cells, is an essential cellular process in both development and physiology; defective efferocytosis would lead to inflammation. In this interesting MS by Schappe et al, the authors identified TRPM7 as a pH-activated Ca^{2+} -permeant cation non-selective channel in the phagosomal membrane that is required for phagosomal acidification and maturation. A multi-disciplinary approach was used, including fluorescence imaging, electrophysiology, Ca^{2+} imaging, and mouse genetics. In addition, several elegant assays were developed to study efferocytosis, including time-lapse ratiometric imaging of phagosomal pH. The paper is well written, and the results were carefully analyzed and presented. Before the paper is accepted for publication in Nature Communications, following points should be addressed.

- 1) Because TRPM7 appeared to only mediate partial Ca^{2+} signals during phagosomal maturation, to strengthen their hypothesis that TRPM7 functions as phagosomal Ca^{2+} release channel, the authors may consider to use an overexpression approach to boost the TRPM7-dependent Ca^{2+} signals during phagosomal maturation. A caveat is that even the overexpressed TRPM7 proteins do not seem to accumulate highly on the limited membranes of nascent phagosomes (see Fig. 5c).
- 2) The authors used both genetic (e.g., TRPM7 KO) and pharmacological (e.g., FTY720) approaches to demonstrate the essential roles of TRPM7 in phagosomal Ca^{2+} release and acidification. Your conclusions could be made much stronger if both two approaches are used in combination in the same experiments. Based on Fig. 5f, FTY720 can rapidly inhibit TRPM7 currents. Can you show the acute effects of FTY720 on Ca^{2+} signals in WT, but not TRPM7 KO cells?
- 3) Is extracellular Ca^{2+} the source of the luminal Ca^{2+} in the nascent phagosomes? If so, would you see differences in phagosomal Ca^{2+} release in the presence (2 mM) or absence (0 mM) of external Ca^{2+} (see Fig. 6b) ?
- 4) The mechanism by which TRPM7 contributes to phagosomal acidification should be discussed. For example, could pH activation of Na^{+} current provide counterion compensation to sustain V-ATPase-mediated H^{+} influx?

RESPONSE TO REVIEWER COMMENTS

We thank the Reviewers for their time and thoughtful suggestions. The Reviewers' comments greatly improved our manuscript, and their enthusiasm for the study design and impact on the field are appreciated. We addressed all reviewer concerns, detailed point-by-point below, including new figures/panels of data and textual revisions to improve the quality of our manuscript.

Reviewers' comments:

Reviewer #1 (Remarks to the Author):

Unclear are the spatiotemporal calcium responses of apoptotic engulfment by macrophages, in contrast to studies of Fc-receptor mediated phagocytosis. Stimulated bone marrow-derived macrophages largely fail to acidify phagosomes when incubated with fast calcium chelator BAPTA. This is in contrast to slow Ca-chelator EGTA-AM and appears independent of basal modulation of lysosomal function. This is a resourceful approach to tackle the challenging question of calcium flux in macrophages and consistent with the careful conclusion with caveats that Ca dynamics proximal to the phagosome are important during efferocytic engulfment.

Thank you for the kind words. Measurement of periphagosomal Ca^{2+} dynamics is indeed a very challenging problem. Our approach of generating transgenic mice that express GCaMP6 in macrophages is the best possible approach currently available. This is a major improvement upon using Ca^{2+} -dyes as well as ectopic expression of GECI. Ideally, we would like to target the GECI only to the efferophagosome but at this time such an approach is not feasible because the proteomic characterization of efferophagosomal membrane has not been accomplished.

GCaMP6s-CX3CR1-Cre mice in support exhibit Ca^{2+} oscillations during initial stages of apoptotic cell engulfment, and TRPM7 is newly implicated after a selective screen, which was validated by percent knockdown of candidate genes. TRPM7 is required for efferocytic acidification.

Yes, identification of TRPM7 as a major regulator of efferophagosome acidification and periphagosomal Ca^{2+} dynamics is indeed the main thrust of this manuscript.

It is laudable that the Investigators seek an *in vivo* test of the findings in culture, although the experiment relies on the infusion of apoptotic cells that were prepared in culture. Thus, is this really *in vivo* efferocytosis?

Our approach aims to evaluate clearance of dead cells in a physiological environment, and we concede that this is not the same as evaluating actual efferocytosis of native apoptotic cells in the context of the organ architecture and physiology. But note that we have deleted TRPM7 only in myeloid cells and in physiological setting apoptotic cells are efferocytosed by a multitude of cell types albeit at different efficiencies. Dissecting the role of TRPM7 through such complexity would be very difficult. In the experimental design we chose, we take advantage of the fact that myeloid cells invade the peritoneal cavity very efficiently. In the revised manuscript, we clarify these distinctions (lines 195 to 199), and we thank the reviewer for highlighting this important point.

This Reviewer also appreciates the steps to control basal effects of the *Trpm7* knockout model, however the F4/80 Cd11c analysis of the peritoneal population is a relatively simplistic analysis of the heterogeneous peritoneal cell subset population. Another trivial explanation of the results in this model could potentially be ascribed to effects on expression of efferocytic receptors, which could potentially be resolved by examining levels of apoptotic to phagocyte cell binding.

We selected an experimental model that allowed for the measurement of tissue resident macrophages and myeloid cells *in situ* and utilized mice with a conditional deletion of *Trpm7* in myeloid phagocytes. This

enabled experiments on a comparable temporal scale to *ex vivo* validation, mitigated potential artifacts of tissue damage at the measurement site (e.g. air pouch), and avoided off-target effects of pharmacological models (e.g. clearance of dexamethasone-induced dead cells). Previous studies report peritoneal macrophage heterogeneity in the context of LPS/thioglycolate-elicited cells, which relies on high/low expression of canonical macrophage markers CD11b and F4/80 (PMID: 20133793). Our *in vivo* model measured efferocytosis by basal populations of peritoneal macrophages using an inclusive classification scheme (CD11b+; F4/80+) that did not rely on demarcation of high/low marker expressing cells or elicitation with exogenous adjuvants. We did not observe differences in frequency of cargo-associated peritoneal macrophages (**Fig 3b**), and TRPM7 KO macrophages exhibited increased association with apoptotic cells (**Fig 3c**), despite a major decrease in cargo acidification – this suggests that cargo engagement and phagocyte binding to cargo is not regulated by TRPM7.

Mechanisms of TRPM7 are investigated and data indicate that *Trpm7* is required for phagolysosome fusion. Interestingly, TRPM7 associates with the nascent phagosome and is activated by low pH. Efferocytosis requires TRPM7 for phagosome-proximal elevations in calcium.

Thank you for your reviewing our manuscript and your helpful comments.

Reviewer #2 (Remarks to the Author):

This study reports that the TRPM7 channel promotes the acidification and maturation of macrophage phagosomes during clearance of apoptotic cells, a process known as efferocytosis. Screening for candidate channel genes impacting phagosomal pH, the authors identify TRPM7 as a positive regulator of phagosome acidification. Bone-marrow derived macrophages (BMDM) from TRPM7-deficient mice failed to acidify and had reduced proteolytic activity. FLAG-TRPM7 expressed in a macrophage cell line was detected near phagosomes. Outwardly-rectifying currents were recorded in BMDM from WT exposed to acidic solutions that were absent in BMDM from TRPM7-deficient mice, which exhibited reduced Ca²⁺ elevations around phagosomes. The authors conclude that the TRPM7 channel sustains efferocytosis in macrophages by mediating the flux of sodium and calcium ions from the phagosome to the cytosol. They propose that the efflux of sodium ions provides a counter-current that sustains proton pumping by the V-ATPase while the efflux of calcium generates vicinal calcium elevations that promote phagosome maturation.

Comments: The study is timely, the experiments well-designed, and the data for the most part convincing. The pH recordings with CypHer5E clearly show that TRPM7 is required for phagosomal acidification. The Ca²⁺ recordings in primary macrophages from mice expressing the genetically encoded calcium indicator GCaMp6 show that TRPM7 contributes to cellular Ca²⁺ signals during phagocytosis. On the other hand the electrophysiological and morphological evidences are less solid and would require further confirmation. I also have issues with the presentation and interpretation of some data.

Thank you for the encouraging points. In the revised manuscript, we have further solidified the electrophysiological and morphological findings (described below).

Specifically:

1. The electrophysiological data appear preliminary. Fig. 5F shows recordings from 5 cells, while Fig. 5G and 5H show single recordings. No statistics are provided and the number of cells recorded is not mentioned. Was only one cell recorded for each genotype? These data need to be substantiated with additional recordings and statistically evaluated.

We have greatly improved on these data by increasing the sample size of the recordings. Please note that these currents were recorded in perforated patch configuration of primary macrophages – something that has never been done before. Achieving good seals with primary macrophages is not trivial even for whole cell currents but for recordings using the perforated patch, we need an average of 30-45 minutes per cell. Often, the seals are lost midway, or electrical access is not achieved. Nevertheless, we have persevered and increased the sample size. The reported data are robust and statistically significant.

The interpretation of the patch-clamp data is also questionable. The recordings show outwardly-rectifying currents reversing at 0 mV in Na⁺/K⁺-based solutions, indicating that the current is carried by monovalent cations (predominantly, K⁺ efflux). Yet, the authors conclude that the channel contributes to the periphagosomal Ca²⁺ elevations by conducting Ca²⁺ ions from the phagosome to the cytosol. This claim is not supported by the electrophysiological evidence, and the directionality of the flux cannot be ascertained without knowledge of the phagosomal membrane potential and of the intraphagosomal Ca²⁺ concentration.

We concede that we do not have direct evidence of efferophagosomal Ca²⁺ flux through TRPM7, but we would like to clarify any possible misinterpretations regarding these electrophysiological data. In the perforated patch recordings, electrical access is achieved through the Nystatin-generated pores in the membrane. Nystatin pores are only permeable to monovalent cations (PMID: 1151324, 7534361) – the recorded current is therefore indeed carried by monovalent cations. These recordings are not designed to capture precise physiological conditions but to demonstrate pH-dependent activation of TRPM7. In the efferophagosome membrane, the directionality of I_{TRPM7} will be dictated by the electrochemical gradients across that membrane. The ionic composition of the nascent efferophagosome is likely to be very similar to the extracellular space – as H⁺ are injected, the membrane will tend to get hyperpolarized. In such conditions, our understanding of TRPM7 biophysics, which is well-studied, dictates that there will a net cationic I_{TRPM7} from the phagosome into the cytosol. Even a strictly monovalent current through TRPM7 has important implications because it provides the countercurrent to sustain the injection of H⁺ into the maturing phagosomes. At this point, we cannot be certain that TRPM7 is only activated by acidification – there may be other activating mechanisms as well. It is exceptionally challenging to resolve all of these possibilities in any single study.

2. The evidence that TRPM7 is present on the membrane of phagosomes is inconclusive. TRPM7 is known as a plasma membrane channel. Here the authors are proposing that TRPM7 is acting as a phagosomal calcium channel. Critical for this claim is to show that TRPM7 actually localizes to the phagosomal membrane, but the confocal images presented are unconvincing. Fig 5C and S4B show immunostaining of LR73 phagocytes expressing FLAG-tagged TRPM7, but the images do not allow a proper evaluation of the labelled structures. Higher resolution and magnification images should be provided, after deconvolution, and the authors should make the complete stack available. Images from the stream analysis should be provided as it is impossible to judge whether that analysis is convincing without seeing the images. A control staining on untransfected cells should be provided. Immunostaining in macrophages are notoriously difficult, as in addition to abundant Fc receptors that bind to IgG, they contain a lot of vesicles that are highly autofluorescent, especially if cells are cultured in media containing phenol red. It is unclear if some of this staining is non-specific. Pre-treating the cells with 0.1% sodium borohydride can quench autofluorescence if the authors find this is a problem. If the difference is not striking it might be good to run these untransfected but stained cells through the image stream to see how specific this colocalization signal really is. There may be an increase in vesicle clustering around phagosomes that is TRPM7-dependent, but from an action at the PM and not at the phagosome membrane. The blotchy staining in the middle of the cell could be vesicles as suggested but could also represent TRPM7 in the ER. ER cisternae at membrane contact sites with phagosomes would not be distinguishable from the phagosomal membrane at this level of resolution. Double staining with PM markers such as Cell Mask, ER marker like ER tracker, and phagosomal membrane marker

such as GFP-lamp1 would document whether the TRPM7 immunoreactivity is really co-localizing with the phagosomal membrane.

These are valid points and we have worked hard to address these concerns. The protein levels of TRPM7 in any cell type are so low that it is not detectable in the protein lysates without immunoprecipitation. None of the available antibodies (including two generated in our own lab) detect native TRPM7 in immunocytochemistry. We even generated a gene-edited mouse to address this without much success. Given these technical limitations, we can only offer evidence based on the immunocytochemistry of ectopically expressed TRPM7. We employed the suggestions offered above and performed new experiments (**Fig 5c**), and these provide more substantial evidence showing that FLAG-TRPM7 is localized proximal to the phagosome and in a membrane compartment distinct from the endoplasmic reticulum (discussed further below). As suggested by the Reviewer, this new image is provided as a complete image stack (**Supplementary File 1**), rendered as a movie (**Supplementary Movie 2**), and additional images are provided in **Supplementary Fig. 4**. This new experimental data clearly shows cells with and without expression of FLAG alongside an endogenous endoplasmic reticulum protein expressed in all cells. We note that all images contain cells successfully transfected with FLAG-TRPM7 expression and some neighboring cells which do not express FLAG-TRPM7 – these serve as internal negative controls within each image.

Briefly, as shown in **Fig. 5c** and discussed in the manuscript, we observe FLAG-TRPM7 expression at both the plasma membrane, vesicles, and the nascent phagosome. FLAG-TRPM7 is localized to subcellular regions distinct from the endoplasmic reticulum (marked by anti-PDI), indicating that TRPM7 localization is not an artifact of ER-accumulated unfolded protein aggregation. Our observed expression pattern is also consistent with published ectopic localization experiments in other cell types (PMID: 28696294, 17088214). We conclude that TRPM7 is localized to the phagosome, but whether TRPM7 is recruited from the plasma membrane or endosomes remains unclear at this point. That aspect is a subject of an independent study focused specifically on TRPM7 trafficking in macrophages.

To address technical concerns of autofluorescence, we performed new experiments using sample treatment with 0.1% sodium borohydride (Reviewer's suggestion; shown in **Fig. 5c** and described in *Methods*) and a commercial reagent (TrueBlack; Biotium) that targets granule autofluorescence. We observed decreases in autofluorescence between treated and untreated samples, but we do not observe difference in the quantitative or qualitative results in new immunostaining experiments (describe further below), as FLAG+ phagocytes are clearly distinguished from surrounding cells. We greatly appreciate the Reviewer's suggestion for membrane and organelle marker reagents; however, these are incompatible with sample fixation and permeabilization required for intracellular FLAG immunostaining. We performed new experiments with CellMask and ER tracker according to the manufacturer's recommended protocols, but fluorescent labeling was poorly retained by fixed and permeabilized samples (4% PFA with 0.1% Triton X-100 and 0.05% Tween-20). To address whether ectopically expressed FLAG-TRPM7 co-localized with endoplasmic reticulum, we immunostained for FLAG and PDI, a robust endoplasmic reticulum marker protein [PMID: 15158710, 16815710] (new images are shown in **Fig 5c** and **Supp. Fig 4d-e**). As expected, we observed co-localization of ER and FLAG in peri-nuclear domains, consistent with ectopically expressed protein, but there was minimal co-localization observed proximal to the phagosome.

As suggested, we also incorporated images of cells from ImageStream flow cytometry experiments (**Fig 5b** and **Supplementary Fig. 5**). Ectopic expression of TRPM7 in RAW 264.7 cells increased cargo uptake and acidification relative to control cells which expressed GFP. The ImageStream instrument (Luminex) is a multispectral flow cytometer that simultaneously acquires microscopy images of acquired cells as they pass through the flow cell. This approach enables quantitative image analysis of thousands of cells, rather than selected image analysis of coverslips. The acquired image resolution (0.1 to 0.25 μm^2 pixel area, 128 μm field of view) is suitable for measuring engulfment by phagocytes. FLAG-TRPM7 co-localized with engulfed cargo to a greater extent than cytosolic GFP (**Fig 5b**), demonstrating a preferential phagosome-

proximal localization of TRPM7, rather than non-specific organization, during phagocytosis. Of course, we realize that ImageStream does not yield a thin optical section like confocal microscopy but averaged over thousands of cells, it offers a probability of colocalization. For clarity, we updated the description of this technique and analysis in the *Results* (lines 275 to 288) and *Methods* (lines 663 to 681) of the revised manuscript.

3. The DQ-Green unquenching assay does not report phago-lysosome fusion, as incorrectly stated in Fig 4A and in the text on pp. 10-11. Instead, the assay reports the proteolytic capacity of phagosomes, which is greatly influenced by the ambient pH. The difference observed could thus reflect a reduced activity of acidic proteases in TRPM7 KO cells that fail to acidify. To conclude that decreased fusion of lysosomes is actually also occurring one would need to use a FRET-based approach as in Russell and colleagues (<https://doi.org/10.1002/0471142735.im1434s102>). At such, the DQ-BSA assay reflects decreased proteolytic activity, not defective fusion, and its interpretation should be revised accordingly.

Yes, this is a good point and we have revised **Fig 4a**, *Results* (lines 235 to 264), and associated figure legends to more accurately describe what is being measured— i.e. phagosome proteolysis, not phago-lysosome fusion. As noted by the Reviewer, our results in **Fig 4** clearly show decreased phagosomal proteolytic activity in *Trpm7*-deficient macrophages. Likewise our experiments show, TRPM7 is not required for basal maintenance of lysosomal pH (**Supplementary Fig. 3a**), indicating that TRPM7 regulates phagosome maturation during dynamic acidification of the phagosomal complex.

4. The authors previously showed that TRPM7 mediates an influx of Ca²⁺ at the plasma membrane of macrophage that is essential for their activation by LPS and for the endocytosis of TLR4 <https://www.ncbi.nlm.nih.gov/pubmed/28130783>. Difference in the activation states of BMDM from WT and KO mice (cultured for 7 days in FBS and conditioned media) could therefore contribute to the phenotype observed. Reduced macrophage activation and endocytosis could both impact the phagosome maturation process during efferocytosis. A reduced influx of Ca²⁺ across the membrane could contribute to the reduced Ca²⁺ signals recorded in macrophages from TRPM7 cells during efferocytosis. These points should be discussed and the contribution of the Ca²⁺ entering across plasma membrane channels should be evaluated by adding channel blockers after particle engulfment.

When cultured in high quality endotoxin free FBS, we have not observed evidence of differential “activation states” in *Trpm7*^{-/-} macrophages and is therefore highly speculative. There is already considerable evidence in the revised manuscript that the deficiency in efferocytosis is not a result of basal differences in activation states of *Trpm7*^{-/-} BMDMs: Outside the context of efferocytosis, lysosomal pH (**Supplementary Fig 3a**), ATP-elicited Ca²⁺ dynamics (**Fig 6b**), and recovered numbers of bone-marrow cells and cultured BMDMs are comparable between wild type and *Trpm7*^{-/-} cells. Furthermore, the initial observation came from siRNA knockdown experiments conducted in wild-type BMDMs (**Fig 1e-f** and **Supplementary Figure 2**). Likewise, please note that blocking TRPM7 channel activity in wild-type BMDMs (with FTY720) recapitulates the phenotype (**Fig. 5d**), although FTY720 may have other off-target effects. Comparing primary macrophages obtained from wild type and gene-targeted mice has its caveats, but such studies are still a significant improvement over cultured cell lines.

With regard to plasma membrane Ca²⁺ channels, a parsimonious model would be for plasma membrane channels, which facilitate SOCE and refilling of ER stores, to augment engulfment and phagosome maturation. Previous studies (PMID: 26109647) and this manuscript (**Supplementary Fig 2**; discussed lines 464-491) clearly report that macrophages that lack SOCE do not display functional defects in phagocytosis. Plasma membrane Ca²⁺ channels may also regulate domains of membrane recycling to support initial engulfment or coordinate assembly of *de novo* protein complexes during phagocytosis – this remains an exciting, but speculative, area for future studies.

Other points:

5. The use of quiescent cells in Figs 6 and S5 as a reference to normalize Ca²⁺ responses occurring in phagocytosing cells is questionable. First, it is not clear what this representation adds compared to the phagosome/cytosol ratio. Second, as discussed above, WT and TRPM7 null cells could have different activation states or different Ca²⁺ influx rates, resulting in different resting Ca²⁺ levels. Finally, the degree of Ca²⁺ activity would be expected to be correlated to the phagocytic activity, yet cells with different phagocytic indexes are grouped in the “engulfing” condition. It would make more sense to compare Ca²⁺ signals occurring in cells ingesting a comparable number of phagosomes. The fluorescence images and recordings should be expressed as F/F₀.

We would like to re-emphasize that we have not seen any significant differences in the activation states of wt and *Trpm7*^{-/-} BMDMs. As shown in Figure 6b, the Ca²⁺ responses to purinergic signals are not significantly different, nor presumably lysosomal acidification of lysosensor molecules (Supplementary Figure 3a). With regard to **Fig 6** and **Supplementary Figure 6**, we expanded our description for clarity of quantification in *Methods* (lines 661 to 670) and provide a procedural vignette in Supplementary Fig 6a-c. “F/F₀” refers to changes in relative fluorescence, and we report changes in phagosomal-proximal fluorescence relative to the cytosol and between engulfing and non-engulfing cells. For each image, ten ROI from at least 3 non-engulfing cells were used to establish baseline (“non-engulfing” fluorescence at each time point) – this internal control minimized coverslip-to-coverslip and timepoint variability which can be introduced by comparing fluorescence across samples with non-ratiometric Ca²⁺ indicators, including GCaMP6. Throughout other experiments (Fig 2, 3, and 4) and in Fig 6, we observed comparable association with apoptotic cell cargo between wt and *Trpm7*^{-/-} BMDMs. It is interesting to speculate how phagocytic indices might “scale” relative to the phagocyte (macrophage vs. ‘non-professional’) and Ca²⁺ dynamics – this may underlie different phagocyte capacities or vary based on cargo, but this is beyond the scope of our study.

6. Fig S1H: The apparent increase in cargo association in the BAPTA condition could reflect the pH-dependent fluorescence of the Cell Trace Violet, illustrated in Fig S1D. Please discuss.

We have revised our discussion of this point (lines 82-91 and 108-115). Fluorescence of CellTrace Violet-labelled cells remains robust across a range of pH levels (pH 7.4 to 4.5). Although CellTrace fluorescence decreases at pH <4.5, cargo-associated phagocytes are still identifiable as CellTrace fluorescence remains orders of magnitude brighter than phagocyte autofluorescence. BAPTA-AM pre-treated phagocytes exhibited increased CellTrace fluorescence, which may be due to increased accumulation of cargo or inhibition of phagosome maturation-dependent acidification - both would effectively increase the measured CellTrace fluorescence.

7. The group of Reinhold Penner showed that TRPM7 regulates SOCE through its kinase domain, and that Ca²⁺ influx through TRPM7 is essential for the refilling of ER Ca²⁺ stores <https://www.ncbi.nlm.nih.gov/pubmed/28130783>. The data of Fig 6B are at odd with this finding, since they show normal long-lasting Ca²⁺ responses to an agonist in Ca²⁺ containing media. Did the authors attempt to compare the store-operated Ca²⁺ entry in WT and TRPM7 null cells? In any case, the divergence with earlier studies should be discussed.

We take the findings from Penner group seriously and respect their usual rigor, but these studies were not done in primary macrophages. We explored this possibility but these findings are not recapitulated in primary macrophages. We have compared SOCE responses in wt and *Trpm7*^{-/-} macrophages rather carefully, and in response to multiple triggers, but we see no significant differences. Even in this manuscript (Fig 6b), note that ATP-induced SOCE (triggered primarily by Gq-couple P2Y receptors) is not significantly different. We discuss the relationship between TRPM7 and SOCE in efferocytosis in the revised manuscript (lines 464-499)

8. Pg 9/Fig. 2C. The authors claim there are no significant differences in cargo uptake, but in the figure, there are stars above two of the three pairs of data points indicating significant differences. Is this a small but significant difference? Could the authors please clarify? What p value stars correspond to should be mentioned in the figure legend

We appreciate the attention to this valid point – there is indeed a modest difference in engulfment. As described in *Methods*, we revised our statistical analysis to appropriately apply two-way ANOVA statistical analysis to experiments through the manuscript, including corrections for multiple comparisons (see **Fig 2c, 4c, and 5b**). As discussed in *Results* for **Fig 2c**, there is a profound decrease in cargo *acidification* during efferocytosis by *Trpm7*-deficient macrophages (nearly two-fold relative to WT). Although *Trpm7*-deficient macrophages exhibit a statistically significant decrease in engulfment, we note that this effect is biologically modest (less than a 10% decrease) and is likely an ancillary effect of flux deficits in downstream cargo acidification and digestion. Appropriate revisions have been made in the *Results* and *Discussion*.

Minor comments:

1) Pg 6 End of first paragraph – Fig. 1b should be referred to after describing the Cyt D experiment

Noted, thank you!

2) Pg 8 – I was surprised to see that TRPC1, TRPC3, and TRPC6 were not included in the list of calcium permeable channels in macrophages as several studies have been published about these channels in this cell type. Please comment.

According to the Immgen database, none of these channels are expressed at appreciable levels in macrophage populations and thus were excluded from our initial analysis. In separate experiments, we also attempted to validate the reported expression of TRPC channels in murine macrophages. In BMDMs, we can just barely detect TRPC1 by qPCR but not by immunoblots (in contrast to published studies). Furthermore, native TRPC1 currents have never been reliably isolated in primary cells, and we also failed to do so in macrophages. In the absence of strong evidence to validate macrophage TRPC channel expression and function, we did not pursue this avenue of investigation. .

3) Pg 8/Fig1h – according to journal policy the full data set should be available - a supplementary table would be a useful way to present these data.

Revealing other hits at this stage undermines other ongoing projects in the lab and is not in the interest of trainees toiling away at these projects. The identity of other hits is not a necessary component of this manuscript. Nevertheless, we can discuss this with the editor. One possibility is to simply take this figure out and start the manuscript with the finding that siRNA knockdown of TRPM7 results in the efferocytosis defect.

4) Pg 8/Supp Fig 2d/ ionomycin experiments – data should be shown, even if just in supplementary according to journal policy

We have included this data in the manuscript. As discussed in line 167-170, interestingly, the Ca^{2+} ionophore Ionomycin inhibited phagosome acidification. This suggests that a sustained, global influx of Ca^{2+} is detrimental to phagosome maturation; alternatively, Ionomycin may provide a conduit for phagosomal H^+ efflux to inhibit maturation (PMID: 8061216).

5) The authors need to mention that the FTY720 compound used here to inhibit TRPM7 is also a potent agonist of S1P G-protein coupled receptors.

Yes, this is now mentioned (lines 313 to 315).

Thank you for your comments and suggestions!

Reviewer #3 (Remarks to the Author):

Efferocytosis, phagocytic clearance of apoptotic cells, is an essential cellular process in both development and physiology; defective efferocytosis would lead to inflammation. In this interesting MS by Schappe et al, the authors identified TRPM7 as a pH-activated Ca^{2+} -permeant cation non-selective channel in the phagosomal membrane that is required for phagosomal acidification and maturation. A multi-disciplinary approach was used, including fluorescence imaging, electrophysiology, Ca^{2+} imaging, and mouse genetics. In addition, several elegant assays were developed to study efferocytosis, including time-lapse ratiometric imaging of phagosomal pH. The paper is well written, and the results were carefully analyzed and presented. Before the paper is accepted for publication in Nature Communications, following points should be addressed.

Thank you for the kind words.

1) Because TRPM7 appeared to only mediate partial Ca^{2+} signals during phagosomal maturation, to strengthen their hypothesis that TRPM7 functions as phagosomal Ca^{2+} release channel, the authors may consider to use an overexpression approach to boost the TRPM7-dependent Ca^{2+} signals during phagosomal maturation. A caveat is that even the overexpressed TRPM7 proteins do not seem to accumulate highly on the limited membranes of nascent phagosomes (see Fig. 5c).

Overexpression studies in BMDMs have been impossible because the transfection efficiency is very low. Overexpression of TRPM7 in RAW264.7 augments cargo association and acidification, relative to expression of GFP-transfected controls (**Supplementary Fig 4a**). We agree with the caveat noted by the reviewers, as overexpressed TRPM7 may not traffic appropriately. Generating TRPM7 overexpressing stable cell lines has also proved to be extremely difficult because chronic overexpression appears to be toxic.

2) The authors used both genetic (e.g., TRPM7 KO) and pharmacological (e.g., FTY720) approaches to demonstrate the essential roles of TRPM7 in phagosomal Ca^{2+} release and acidification. You conclusions could be made much stronger if both two approaches are used in combination in the same experiments. Based on Fig. 5f, FTY720 can rapidly inhibit TRPM7 currents. Can you show the acute effects of FTY720 on Ca^{2+} signals in WT, but not TRPM7 KO cells?

Our study employs primary mammalian cells studied under well-established culture conditions with Cre-lox-mediated deletion of *Trpm7*. FTY720 is well-studied agonist of S1P-receptors. The lack of specificity for TRPM7 introduces several caveats which make additional effort in this direction less attractive.

3) Is extracellular Ca²⁺ the source of the luminal Ca²⁺ in the nascent phagosomes? If so, would you see differences in phagosomal Ca²⁺ release in the presence (2 mM) or absence (0 mM) of external Ca²⁺ (see Fig. 6b) ?

At this point, we are assuming that the main source of phagosomal Ca²⁺ is extracellular Ca²⁺. The main problem with the suggested experiment is that engulfment itself is affected in 0 mM extracellular Ca²⁺ and this confounds the interpretation.

4) The mechanism by which TRPM7 contributes to phagosomal acidification should be discussed. For example, could pH activation of Na⁺ current provide counterion compensation to sustain V-ATPase-mediated H⁺ influx?

We have discussed the proposed model in the revised manuscript (please see lines 33-60, 409-437).

REVIEWER COMMENTS

Reviewer #1 (Remarks to the Author):

no further comments

Reviewer #2 (Remarks to the Author):

The authors have performed substantial experimental work in this new submission. They provide additional data, clarifications, and discussions that solve all the issues raised in the first round of review. The study is now a solid piece of work that will likely set new standards in the field. I've only two minor remarks.

In figure 5F the legend states: "Time-current relationship of ITRPM7 activation and block by FTY720" but the traces show current activation by acidic pH and block by alkaline pH not by FTY720.

In the rebuttal the author claim that only cationic currents can be recorded in the perforated patch configuration because the pores generated by nystatin only allow the flux of monovalent cations. This is incorrect. Nystatin pores provide electrical conduction (via monovalent cations) between the pipette and the cell cytosol, but the whole-cell currents recorded flow across plasma membrane channels and can be carried by any ion that can permeate the membrane of the recorded cell.

Reviewer #3 (Remarks to the Author):

This remains to be an interesting study. However, it is not clear whether the technical difficulties are sufficient to preclude the authors from conducting some of the suggested experiments. For instance, if BMDMs were too difficult to be transfected using conventional methods, can you perform the test in RAW264.7 cells? Likewise, Ca²⁺ imaging is a relatively -specific assay, if FTY720 can be used to probe the effect of TRPM7 on phagosomal acidification, why is it not possible to investigate its effects on Ca²⁺ release? Finally, the effect of low pH on TRPM7 appeared to be potentiating outward currents in the current study, yet in the literature low pH was reported to selectively augment inward currents without affecting the outward currents (see PMID 16009728). Note that low pH is known to activate the

ubiquitously-expressed PAC currents (PMID: 31023925) in the presence of extracellular Cl⁻, a recording condition used in the current study. Note that the proton-activated current in Fig. 5f resembled I-PAC, as opposed to I-TRPM7. Could the residual current in the presence of FTY720 be mediated by PAC, at least in some macrophages?

RESPONSE TO REVIEWER COMMENTS

We thank the reviewers for their constructive comments and appreciate their interest and enthusiasm for our manuscript. Here, we address the last of reviewer concerns with point-by-point responses and the manuscript has also been suitably revised.

Reviewer #1 (Remarks to the Author):

no further comments

We appreciate the time and comments from the referee.

Reviewer #2 (Remarks to the Author):

The authors have performed substantial experimental work in this new submission. They provide additional data, clarifications, and discussions that solve all the issues raised in the first round of review. The study is now a solid piece of work that will likely set new standards in the field. I've only two minor remarks.

Many thanks for the kind words and supportive comments.

In figure 5F the legend states: "Time-current relationship of ITRPM7 activation and block by FTY720" but the traces show current activation by acidic pH and block by alkaline pH not by FTY720.

Thank you for catching that mix up! We have corrected the figure legends for **Figure 5f-h** and **Supplementary Fig 5d**.

In the rebuttal the author claim that only cationic currents can be recorded in the perforated patch configuration because the pores generated by nystatin only allow the flux of monovalent cations. This is incorrect. Nystatin pores provide electrical conduction (via monovalent cations) between the pipette and the cell cytosol, but the whole-cell currents recorded flow across plasma membrane channels and can be carried by any ion that can permeate the membrane of the recorded cell.

Yes – the reviewer is indeed correct! Although Nystatin pores themselves are only permeant to monovalent cations, the configuration does allow the measurement of all currents passing between the bath solution and plasma membrane. We concede the confusion in our previous response letter, but our main point stands that the use of perforated patch was primarily to demonstrate pH dependent activation of TRPM7 without dialyzing the cytosol and possibly depleting the regulatory machinery.

Reviewer #3 (Remarks to the Author):

This remains to be an interesting study. However, it is not clear whether the technical difficulties are sufficient to preclude the authors from conducting some of the suggested experiments. For instance, if BMDMs were too difficult to be transfected using conventional methods, can you perform the test in RAW264.7 cells? Likewise, Ca²⁺ imaging is a relatively -specific assay, if FTY720 can be used to probe the effect of TRPM7 on phagosomal acidification, why is it not possible to investigate its effects on Ca²⁺ release?

Macrophages express multiple pattern recognition receptors that recognize nucleic acids and initiate cell intrinsic responses that degrade these nucleic acids before they enter the nuclei. All myeloid cell lines, including RAW264.7 cells are therefore incredibly resistant to ectopic transection of plasmids, but fare better with shorter siRNA fragments. With plasmid DNA, transfection of RAW 264.7 gives us a transfection efficiency of a <3% - we tried most available transfection reagents, but it is simply difficult to carry out ectopic expression in myeloid cells. With such low efficiency in mind, we designed experiments with bulk transfections to be performed by flow cytometry using large numbers of cells (0.2×10^6 cells per test; line 680 in *Methods*) and analyzed by gating on the population of GFP⁺ cells, which still yields thousands of cells for analysis at this scale. Through such analysis, we show that RAW 264.7 cells transfected with FLAG-TRPM7+GFP exhibited increased cargo association and acidification

relative to GFP-alone transfected controls compared to RAW cells transfected with GFP plasmid (negative control) as measured by flow cytometry (**Supplementary Fig 4a**). We have tried sorting these cells and re-seeding them, but they tend not to adhere and when they do, the morphology is often indicative of ongoing apoptosis. The large coding sequence of TRPM7 also precludes lentiviral based transduction – the insert size between the LTRs is >8kb and it has been impossible for us to derive titers that can transduce myeloid cells (primary or cell lines) with TRPM7 constructs. This is why most reported studies that evaluate the properties of ectopically expressed TRPM7 variants are carried out in simple cell lines such as 293T cells. It is an enduring technical challenge for us. Even for GCaMP6, we had to generate transgenic mice because we failed to efficiently transfect myeloid cells with GCaMP6.

More to the conceptual point, we show that pre-treatment of BMDMs with FTY720 dampens phagosome-proximal Ca^{2+} elevations during efferocytosis (**Fig 6h** and **Supp Fig 5f**). This observation is consistent with FTY720 inhibition of cargo association and acidification during efferocytosis (**Fig 5d** and **Supp Fig 5b and 5c**). While FTY720 may have other off-target effects including agonism of S1P receptors (lines 313 to 315), the Cre-mediated genetic approaches we employed throughout the manuscript provide the strongest evidence possible to establish TRPM7's function in efferocytosis.

Finally, the effect of low pH on TRPM7 appeared to be potentiating outward currents in the current study, yet in the literature low pH was reported to selectively augment inward currents without affecting the outward currents (see PMID 16009728).

Yes, we noticed this discrepancy with literature but note that in the cited study carried out by Dr. Yue's lab used 293T cells ectopically expressing TRPM7, and in whole cell configuration – not perforated patch of primary macrophages. We can reproduce their studies in 293T cells and do not dispute their findings, but we simply do not see this kind of TRPM7 activation in a perforated patch configuration of macrophages. We think that our approach is more principled because it preserves the regulatory environment of TRPM7 and evaluates native TRPM7, not ectopically expressed TRPM7.

Note that low pH is known to activate the ubiquitously-expressed PAC currents (PMID: 31023925) in the presence of extracellular Cl^- , a recording condition used in the current study. Note that the proton-activated current in Fig. 5f resembled I-PAC, as opposed to I-TRPM7. Could the residual current in the presence of FTY720 be mediated by PAC, at least in some macrophages?

We are aware of the PAC currents – it is a very insightful point by the reviewer. The pH activated current we see is inhibited by Mg^{2+} and FTY720 so a TRPM7 component is not in doubt. That said, we have recently established that PAC is expressed in primary macrophages, so there is likely a PAC contribution as well, though expression levels vary across subtypes of macrophages. In an ongoing but different study, we are evaluating the role of PAC in macrophage functions - siRNA mediated knockdown of PAC does not have a significant impact on efferocytosis.

REVIEWERS' COMMENTS

Reviewer #3 (Remarks to the Author):

As the readers may have the same concerns as the reviewers do, the authors may wish to include some of the discussions in their rebuttal letter in the MS. Finally, if I was an author, I would make sure that PAC currents are not seen in TRPM7 KO macrophages.

We appreciate the reviewers for their time, comments, and positive assessment of our study. Here, we address the final reviewer concern with a suitably revised manuscript:

REVIEWERS' COMMENTS

Reviewer #3 (Remarks to the Author):

As the readers may have the same concerns as the reviewers do, the authors may wish to include some of the discussions in their rebuttal letter in the MS. Finally, if I was an author, I would make sure that PAC currents are not seen in TRPM7 KO macrophages.

To highlight this insightful point regarding PAC currents, we update the manuscript discussion with the following text: "Although the pH-activated current was inhibited by Mg^{2+} , FTY720, and genetic deletion of *Trpm7*, other ion channels regulated by pH, such as PAC, likely play important roles in pH-sensing by macrophages."

While TRPM7 KO macrophages would be expected to maintain expression of PAC channels, we appreciate the interest in understanding mechanisms of pH-sensitive channels in macrophage biology, and these mechanisms may also vary by macrophage subtype and tissue context.